🔵 **eLife**

# Latent gammaherpesvirus exacerbates arthritis through modification of age-associated B cells

Isobel C Mouat[1], Zachary J Morse[1], Iryna Shanina[1], Kelly L Brown[2], Marc S Horwitz[1]*

[1]Department of Microbiology and Immunology, The University of British Columbia, Vancouver, Canada; [2]Department of Pediatrics, Division of Rheumatology, and British Columbia Children's Hospital Research Institute, The University of British Columbia, Vancouver, Canada

**Abstract** Epstein-Barr virus (EBV) infection is associated with rheumatoid arthritis (RA) in adults, though the nature of the relationship remains unknown. Herein, we have examined the contribution of viral infection to the severity of arthritis in mice. We have provided the first evidence that latent gammaherpesvirus infection enhances clinical arthritis, modeling EBV's role in RA. Mice latently infected with a murine analog of EBV, gammaherpesvirus 68 (γHV68), develop more severe collagen-induced arthritis and a Th1-skewed immune profile reminiscent of human disease. We demonstrate that disease enhancement requires viral latency and is not due to active virus stimulation of the immune response. Age-associated B cells (ABCs) are associated with several human autoimmune diseases, including arthritis, though their contribution to disease is not well understood. Using ABC knockout mice, we have provided the first evidence that ABCs are mechanistically required for viral enhancement of disease, thereby establishing that ABCs are impacted by latent gammaherpesvirus infection and provoke arthritis.

*For correspondence:
mhorwitz@mail.ubc.ca

Competing interests: The authors declare that no competing interests exist.

## Introduction

Rheumatoid arthritis (RA) is one of the most common autoimmune diseases in adults, though the etiology and pathophysiology are not fully understood (*Humphreys et al., 2013*; *Firestein and McInnes, 2017*). RA, as well as other autoimmune diseases including multiple sclerosis (MS) and systemic lupus erythematosus (SLE), is associated with Epstein-Barr virus (EBV) infection (*Balandraud et al., 2003*; *Ascherio and Munger, 2007*; *Gross et al., 2005*). EBV infection typically takes place during childhood or adolescence, while RA generally becomes symptomatic during middle age, indicating that the latent EBV infection likely modulates the immune system over time in a manner that contributes to the development of RA (*Humphreys et al., 2013*; *Dowd et al., 2013*; *Fourcade et al., 2017*; *Alamanos et al., 2006*). The circulating EBV load is higher in individuals with RA than in otherwise healthy adults (*Balandraud et al., 2003*), and RA patients have increased levels of antibodies specific to multiple EBV-encoded proteins (*Blaschke et al., 2000*; *Ferrell et al., 1981*; *Catalano et al., 1979*; *Alspaugh et al., 1981*; *Hazelton et al., 1987*). Further, RA patients have increased EBV-specific CD8[+] T cells (*Lünemann et al., 2008*), yet these cells have a reduced ability to kill EBV-infected B cells when compared to the same subset of EBV-specific CD8[+] T cells from healthy controls (*Takei et al., 2001*). However, the precise role of EBV in RA pathogenesis remains unknown.

Evidence from in vivo models is scarce and previous studies have focused primarily on the direct relationship between EBV infection and damage to the joint capsule, with little attention given to systemic effects of EBV infection on immune modulation preceding and continuing throughout

**eLife digest** Rheumatoid arthritis is one of the most common autoimmune diseases, leaving patients in pain as their immune system mistakenly attacks the lining of their joints. The precise cause is unknown, but research suggests a link to the Epstein-Barr virus, the agent responsible for mononucleosis (also known as glandular fever). After infection and recovery, the virus remains in the body, lying dormant inside immune 'B cells' which are often responsible for autoimmune diseases. Of particular interest are a sub-group known as 'age-associated B-cells', which are mostly cells left over from fighting past infections such as mononucleosis. Yet, the link between Epstein-Barr virus and rheumatoid arthritis remains hard to investigate because of the long gap between the two diseases: the virus mostly affects children and young people, while rheumatoid arthritis tends to develop in middle age.

To investigate how exactly the two conditions are connected, Mouat et al. created a new animal model: they infected young mice with the murine equivalent of the Epstein-Barr virus, and then used a collagen injection to trigger rheumatoid arthritis-like disease once the animals were older.

Next, Mouat et al. monitored the paws of the mice, revealing that viral infection early in life worsened arthritis later on. These animals also had more age-associated B cells than normal, and the cells showed signs of participating in inflammation. On the other hand, early viral infection did not make arthritis worse in mice unable to produce age-associated B cells. Taken together, these results suggest that the immune cells are required to enhance the effect of the viral infection on rheumatoid arthritis. This new insight may help to refine current treatments that work by reducing the overall number of B cells. Ultimately, the animal model developed by Mouat et al. could be useful to identify better ways to diagnose, monitor and treat this debilitating disease.

disease (*Kuwana et al., 2011*; *LeBel et al., 2018*). Mice with humanized immune systems, namely NOD/Shi-*scid*/IL-2Rγ$^{null}$ mice reconstituted with CD34$^+$ hematopoietic stem cells, that were infected with EBV went on to spontaneously develop erosive arthritis, suggesting a causative role of EBV in arthritis development (*Kuwana et al., 2011*). Related, a serum transfer-induced arthritis model was used to demonstrate that Ly6C$^{high}$ monocytes play a role in transporting murine gammaherpesvirus 68 (γHV68), an EBV homolog, to the synovium (*LeBel et al., 2018*). Our group has previously shown that latent γHV68 infection exacerbates experimental autoimmune encephalomyelitis (EAE) and leads to a disease that more closely resembles MS, with increased demyelination and infiltration of CD8 to cells of the central nervous system in γHV68-infected mice (*Casiraghi et al., 2012*). Critically, this enhancement was specific to γHV68; other viruses, including lymphocytic choriomeningitis virus (LCMV) and murine cytomegalovirus (MCMV), did not lead to enhancement of EAE. Additionally, enhancement took place without changes to autoantibody levels. An in vivo model that recapitulates the temporal and systemic immunological aspects of the relationship between EBV and RA is critical.

To examine the relationship between EBV and RA, we have adapted in vivo models of both. γHV68 is a natural pathogen that is a well-established and widely-used murine model of EBV infection that shares an array of characteristics with human EBV infection, including latent persistence in B cells, viral reactivation from latency, a potent CD8 T cell response, and immune evasion tactics (*Olivadoti et al., 2007*; *Wirtz et al., 2016*). Type II collagen-induced arthritis (CIA) is a commonly used model of RA wherein mice are injected with type II collagen emulsified in an adjuvant. Here, we chose to use C57BL/6J mice due to the extensive past characterization of γHV68 infection in C57BL/6 mice and the numerous knockout (KO) strains available on this background. Multiple strains of mice are susceptible to CIA, including C57BL/6 mice that, despite displaying a less severe disease course than other strains, generate a robust T cell response (*Inglis et al., 2007*; *Campbell et al., 2000*). In C57BL/6 mice, CIA follows a chronic disease course with a sustained T cell response, presence of anti-collagen IgG, and infiltration of inflammatory lymphocytes into the joint capsule (*Inglis et al., 2007*). EBV primary infection generally takes place in childhood or adolescence (*Dowd et al., 2013*; *Fourcade et al., 2017*), and RA can occur at any age, though the mean incidence is in the sixth decade of life (*Myasoedova et al., 2010*). Accordingly, we have infected immunologically and sexually mature 6- to 8-week-old C57BL/6J mice with γHV68 and have induced CIA when the mice were adults at 11–13 weeks old. Here we have shown that C57BL/6J

mice infected with latent γHV68 and induced for CIA develop a more severe clinical course and an altered immunological profile compared to uninfected CIA controls, with expanded CD8$^+$ T cells and Th1 skewing. We have utilized γHV68 infection and CIA induction to investigate mechanism(s) by which EBV contributes to RA, in particular through the modulation of age-associated B cells (ABCs).

The role of B cells in the relationship between EBV and RA is intriguing because B cells contribute pathogenically to RA, and EBV infects B cells and alters the B cell profile (*Marston et al., 2010*; *Hatton et al., 2014*). ABCs are a subset of B cells that are of particular interest as they have been implicated in both autoimmunity and viral infection. When compared to healthy adults, the relative proportion and/or absolute circulating counts of ABCs are elevated in RA patients, a subset of individuals with MS, individuals with SLE, and a subset of people with common variable immune deficiency that displays autoimmune complications (*Adlowitz et al., 2015*; *Rubtsov et al., 2011*; *Thorarinsdottir et al., 2019*; *Wang et al., 2019*; *Claes et al., 2016*; *Wang et al., 2018*; *Zhang et al., 2019*; *Rakhmanov et al., 2009*). ABCs are required for disease development in mouse models of SLE (*Rubtsova et al., 2017*). Also, ABCs are increased during viral infections in mice and/or humans including LCMV, γHV68, vaccinia, hepatitis C virus, HIV, and influenza (*Rubtsova et al., 2013*; *Chang et al., 2017*; *Knox et al., 2017*; *Johnson et al., 2020*). ABCs display an array of functional capacities, including the secretion of anti-viral or autoantibodies, initiation of germinal centers, antigen presentation to T cells, and secretion of cytokines (*Rubtsova et al., 2013*; *Chang et al., 2017*; *Knox et al., 2017*; *Johnson et al., 2020*). It is yet to be examined whether ABCs play a role in viral contribution to autoimmunity. We found that ABC KO mice are unable to develop the γHV68-exacerbation of CIA and therefore act as a mediator between viral infection and autoimmunity.

## Results

### Latent γHV68 infection exacerbates the clinical course of CIA

The development of RA often occurs years after initial infection with EBV when the virus is latent. To mimic this temporal relationship, we infected mice with γHV68, waited 5 weeks for the lytic infection to clear and the virus to establish latency, and induced CIA. Clearance of the acute virus and establishment of latency have previously been shown by plaque assay on spleens collected 35 days post-infection (*Casiraghi et al., 2012*; *Barton et al., 2014*). Following CIA induction, mice were assessed three times per week for redness and swelling in the two hind paws (*Figure 1—figure supplement 1A*), which informed a clinical score for each mouse. We observed that CIA in latent γHV68-infected mice (herein referred to as γHV68-CIA mice) had a more severe clinical course than uninfected mice (herein referred to as CIA mice), as evidenced by consistently higher clinical scores and changes in paw heights throughout the clinical course (*Figure 1A–B*, *Figure 1—figure supplement 1B*). γHV68-CIA mice also developed onset of disease symptoms an average of 7 days earlier than CIA mice, reached a higher score at endpoint, and displayed a higher cumulative score (*Figure 1C*, *Figure 1—figure supplement 1C-D*). In agreement with other research groups (*Campbell et al., 2000*), male and female mice displayed similar clinical scores during CIA, and we also did not observe a sex difference in γHV68-CIA mice (*Figure 1—figure supplement 1E*). As expected, latent γHV68-infected mice (without CIA) did not display any signs of disease (*Figure 1A–B*). Titers of anti-type II collagen autoantibodies (total IgG, IgG1, and IgG2c) were elevated in sera from mice with CIA compared to naive mice without CIA, yet were similar in mice with CIA regardless of infection (*Figure 1—figure supplement 1F–H*). Additionally, we found that inducing CIA in γHV68-infected mice did not impact viral load (*Figure 1—figure supplement 1I*), indicating that γHV68 is not reactivating. These findings are in line with our previous work showing that latent γHV68 infection enhances EAE without influencing autoantibody levels or reactivating γHV68 (*Casiraghi et al., 2012*). These data demonstrate that latent γHV68 infection leads to earlier onset and more severe CIA, though the exacerbation is not due to higher titers of autoantibodies against type II collagen or changes in abundance of particular immunoglobulin isotypes.

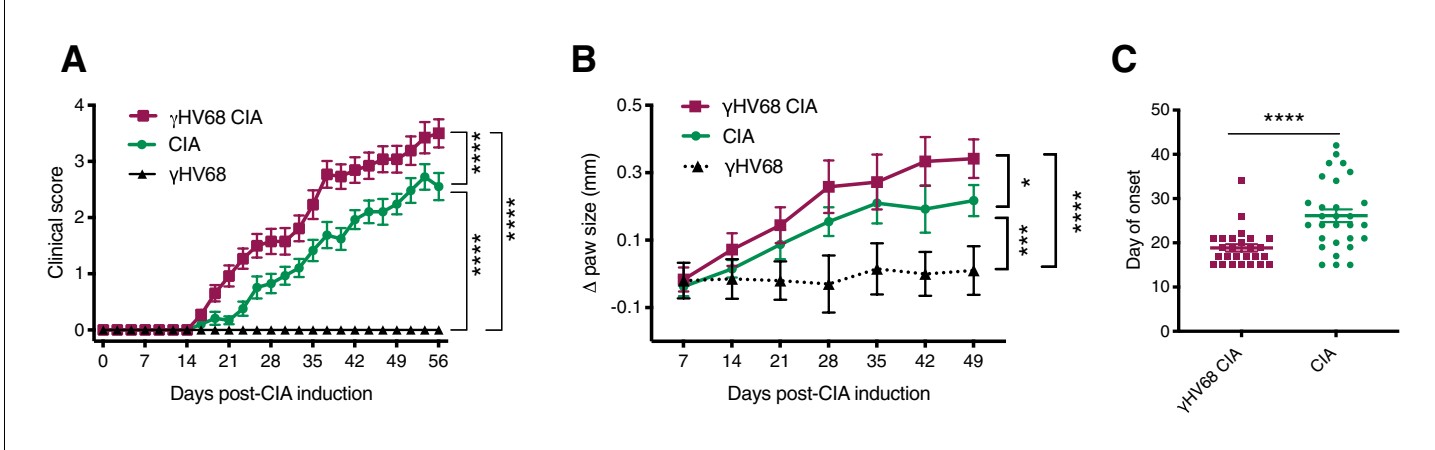

**Figure 1.** Progression of CIA in latent γHV68-infected and control uninfected mice. (**A**) Clinical score (y-axis) of collagen-induced arthritis (CIA) measured three times weekly for 8 weeks (x-axis; days) post-CIA induction in mice without (CIA, filled circles) and with latent γHV68 infection (γHV68-CIA, open squares), and starting at day 35 post-infection in mice infected with latent γHV68 infection but not induced for CIA (γHV68, filled triangles). (**B**) Change (Δ, y-axis) in thickness of hind paws measured with calipers once per week and averaged for each mouse, γHV68-CIA and CIA being measured on the day of CIA induction and γHV68 at day 35 post-infection. (**C**) Day (y-axis) of CIA onset (considered 2 consecutive scoring days of a score of at least 1) in mice (x-axis) without (CIA) and with latent γHV68 infection (γHV68-CIA). Each data point represents an individual mouse. (**A–C**) Data presented as mean ± SEM. Statistical significance determined by (**A, B**) two-way ANOVA with multiple comparisons with F-values 329.22 (**A**) and 17.95 (**B**), (**C**) Mann-Whitney test. ****p<0.0001, ***p<0.001, **p<0.01, *p<0.05. (**A**) n = 10–29 mice per group, four experiments; (**B**) n = 8–20 mice per group, three experiments; (**C**) n = 26–29 mice per group, four experiments.

The online version of this article includes the following source data and figure supplement(s) for figure 1:

**Source data 1.** Progression of CIA in latentγHV68 infected and control uninfected mice source data.
**Figure supplement 1.** Clinical data, autoantibody titers, and viral load.
**Figure supplement 1—source data 1.** Clinical data, autoantibody titers, and viral load source data.

## The profile of immune cells infiltrating the synovium is altered in γHV68-CIA

To assess the types and relative proportions of immune cells infiltrating the joint synovium, synovial fluid cells were collected on day 56 post-CIA induction. Synovial cells were collected from the knee and ankle joints by flushing each joint with phosphate-buffered saline (PBS) and subsequently analyzing isolated cells by flow cytometry. Synovial cells were not collected from naive or γHV68-infected mice without CIA because we would not expect there to be sufficient infiltration of immune cells for analysis. The number of CD8[+] T cells infiltrating the synovium during γHV68-CIA was increased compared to CIA (3.6-fold change), while there was no significant difference in the number of CD4[+] T cells (*Figure 2A*). Additionally, the CD8[+] and CD4[+] T cells in γHV68-CIA synovium displayed a significant increase in Tbet expression compared to those in CIA (*Figure 2B*), indicating Th1 skewing.

As further evidence that infiltrated T cells are immunologically active, we used real-time quantitative PCR (RT-qPCR) to evaluate the expression of key T cell-derived cytokines *Ifng* and *Il17*. The relative expression of *Ifng* in synovium cells of γHV68-CIA mice compared to CIA mice was increased (129-fold change), while the relative expression of *Il17a* was trending down in infected mice (*Figure 2C*; 3.8-fold change), though the sample size was low due to the difficulty of obtaining these samples. Together, these results indicate that IFNγ-producing T cells were preferentially infiltrating the synovium in our model of γHV68-CIA, which is consistent with what was observed in the synovium of RA patients (*Yamada et al., 2008*). Our data also demonstrate a skewing toward cytotoxic CD8[+] T cells in mice latently infected with γHV68 prior to CIA.

## Latent γHV68 infection skews the T cell response toward a pathogenic profile during CIA

To examine how latent γHV68 might contribute to CIA, we specifically examined the systemic T cell profile. It is known that latent γHV68 infection expands cytotoxic T cells and reduces Tregs (*Casiraghi et al., 2015*). Both cell types play a role in CIA with cytotoxic T cells being crucial

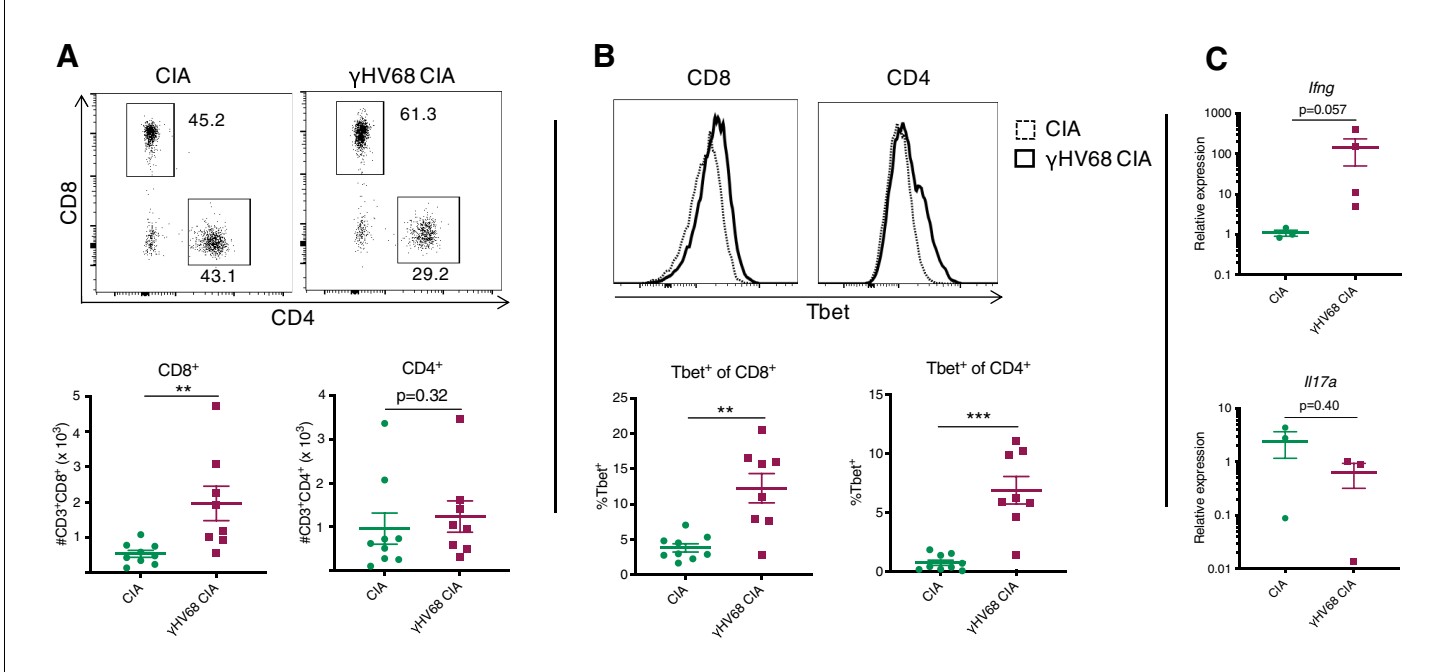

**Figure 2.** Analysis of immune infiltration to synovium between γHV68-CIA and control CIA mice at day 56 post-CIA induction. (A) Representative flow cytometry plots of synovial fluid (SF) CD8+ and CD4+ T cells, previously gated on lymphocytes, singlets, live cells, and CD45+CD3+ cells. Flow cytometry plots (concatenated samples) and graphs of total numbers (y-axis) of CD45+CD3+CD8+ and CD45+CD3+CD4+ T cells in uninfected mice with collagen-induced arthritis (CIA) (filled circles) and γHV68-CIA mice (filled squares). (B) Representative flow cytometry plots of Tbet expression (x-axis) by CD8+ or CD4+ T cells in CIA mice (dotted line) and γHV68-CIA mice (solid line). Samples were previously gated on lymphocytes, singlets, live cells, and CD45+CD3+ cells. Percent of CD8+ and CD4+ T cells positive for Tbet (y-axis, gated on a full-minus-one control) in uninfected mice with CIA (filled circles) and γHV68-CIA mice (filled squares). (C) RNA extracted from synovial fluid cells, reverse transcriptase quantitative PCR (RT-qPCR) performed for *Ifng* and *Il17a*, and relative expression plotted for uninfected mice with CIA (filled circles) and γHV68-CIA mice (filled squares). (A–B) Flow plots are concatenated samples from all CIA or γHV68-CIA samples from an individual experiment, n = 8–9 mice per group; (C) n = 3–4 mice per group; (A–C) one experiment, data presented as mean ± SEM, analyzed by Mann-Whitney test; ****p<0.0001, ***p<0.001, **p<0.01, *p<0.05.
The online version of this article includes the following source data for figure 2:

**Source data 1.** Analysis of immune infiltration to synovium betweenγHV68-CIA and control CIA mice at day 56 post-CIA induction.

mediators of CIA while Tregs play a protective role (*Tada et al., 1996*; *Morgan et al., 2003*). We examined T cells in the spleen and inguinal lymph nodes (ILNs), a draining lymph node in which we observed a significant increase in overall abundance of immune cells during CIA (*Figure 3—figure supplement 1A*). γHV68-CIA mice displayed a decrease in relative proportion of FoxP3+ Tregs and an increase in relative proportion of CD8+ T cells in the spleen compared to control CIA mice (*Figure 3A–B*, *Figure 3—figure supplement 1A, B-C*). This is similar to what was observed in people with RA, as activated CD8+ T cells were increased and Tregs were decreased in the circulation of RA patients compared to otherwise healthy people (*Morita et al., 2016*; *Ramwadhdoebe et al., 2016*). In the ILNs of γHV68-CIA mice, we observed a nonsignificant trend of decreased CD8+ and CD4+ T cell relative proportions, indicating potential T cell egress from the ILNs during disease, and found that the proportion of regulatory T cells was unchanged between CIA and γHV68-CIA mice (*Figure 3A–B*, *Figure 3—figure supplement 1E–G*). We also observed a significant increase in relative proportion of CD11c+CD8+dendritic cells (DCs) in γHV68-CIA mice compared to CIA (*Figure 3—figure supplement 1H–J*). These data show that the T cell profile of γHV68-CIA mice is skewed pathogenically, with decreased Tregs and increased cytotoxic T cells.

## T cell polarization is modulated in γHV68-CIA mice

Although IL17 has been highly studied due to its predominance in animal models of arthritis, both IL17 and IFNγ are involved in RA (*Yamada et al., 2008*; *Ramwadhdoebe et al., 2016*; *Shen et al.,*

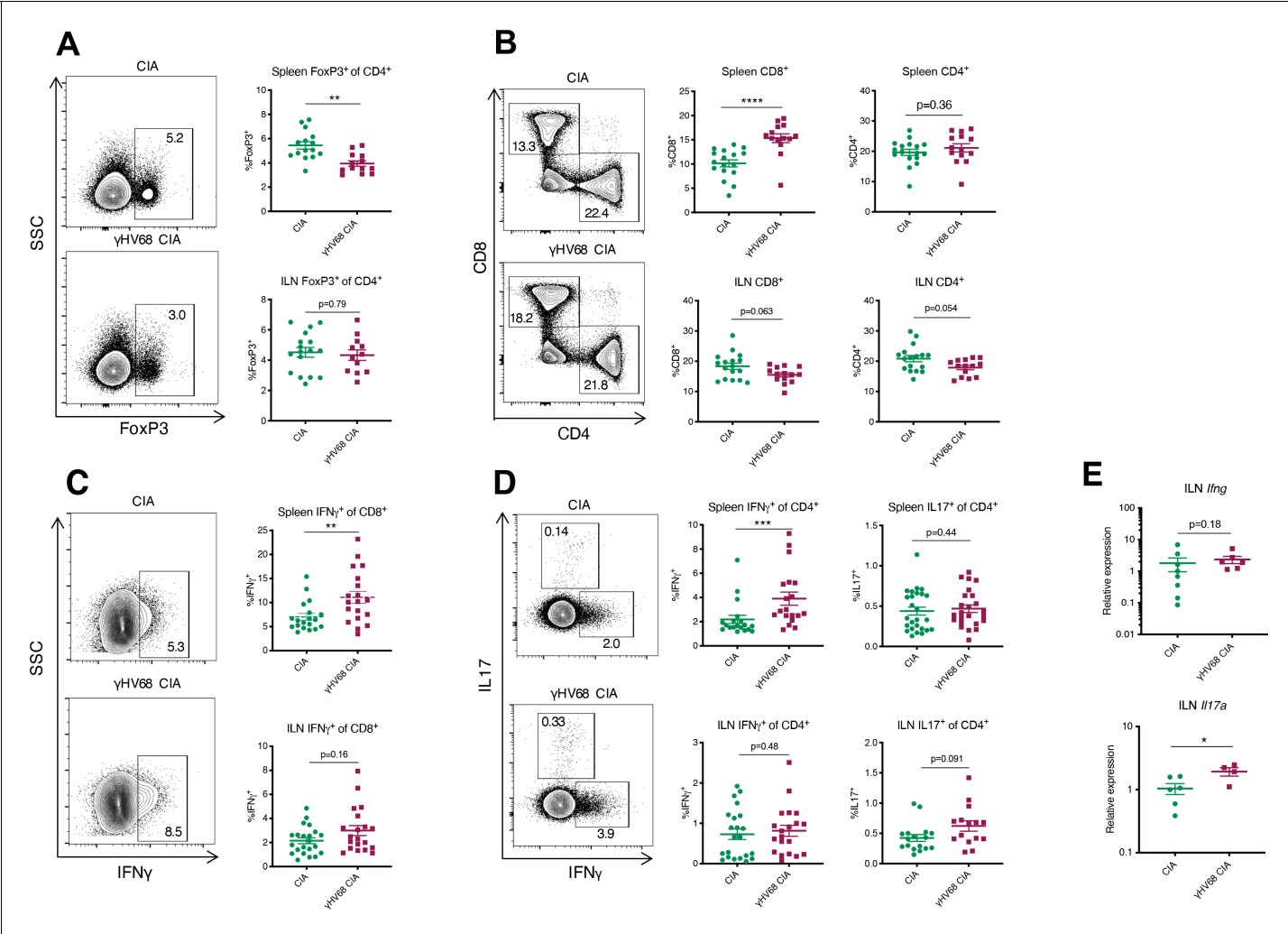

**Figure 3.** Flow cytometry analysis of spleen and ILN T cells at day 56 post-induction of γHV68-CIA and control CIA mice. (A–D) Representative flow cytometry plots of spleen samples previously gated on lymphocytes, live cells, singlets, CD45+CD3+ cells, and (A, D) CD4+ cells, (C) CD8+ cells, of uninfected mice with collagen-induced arthritis (CIA) (upper plot) and γHV68-CIA mice (lower plot). (A, C) Side-scatter (SSC) plotted on the y-axis. (A–D) Percent of immune subsets (y-axis) in the spleens of uninfected mice with CIA (filled circles) and γHV68-CIA mice (filled squares). (A) %FoxP3+ of CD4+; (B) %CD3+CD8+ and %CD3+CD4+ of CD45+; (C) %IFNγ+ of CD8+; (D) IFNγ+ or IL17A+ of CD4+; (E) RNA extracted from inguinal lymph node (ILN) cells, real-time quantitative PCR (RT-qPCR) performed for *Ifng* and *Il17a*, and relative expression plotted for uninfected mice with CIA (filled circles) and γHV68-CIA mice (filled squares). (A) n = 12–17 mice per group, three experiments; (B) n = 14–17 mice per group, three experiments; (C, D) n = 19–22 mice per group, three experiments; (E) n = 4–8 mice per group, one experiment. (A–E) Each data point represents an individual mouse. Data presented as mean ± SEM, analyzed by Mann-Whitney test; ****p<0.0001, ***p<0.001, **p<0.01, *p<0.05.

The online version of this article includes the following source data and figure supplement(s) for figure 3:

**Source data 1.** Flow cytometry analysis of spleen and ILN T cells at day 56 post-induction ofγHV68-CIA and control CIA mice.

**Figure supplement 1.** Total numbers of spleen and inguinal lymph node T cell populations at day 56 post-CIA induction.

**Figure supplement 1—source data 1.** Total numbers of spleen and inguinal lymph node T cell populations at day 56 post-CIA induction.

*2009*; *Nistala et al., 2010*). As expected from our previous work with γHV68-EAE, we found that in γHV68-CIA, greater numbers of splenic CD8+ and CD4+ T cells express IFNγ compared to CIA alone (*Figure 3C–D*). There is a maintenance of Th17 cells in the spleen, with a similar proportion of CD4+ T cells expressing IL17A in CIA and γHV68-CIA (*Figure 3D*). In the ILNs, we observed a significant increase in *Il17a* by RT-qPCR (*Figure 3E*) and a corresponding trend toward more IL17A-expressing CD4+ T cells. We propose that the combined Th1 and Th17 profile observed in γHV68-CIA is more reminiscent of what is observed in people with RA than in CIA without γHV68 infection.

## Latency is required for the clinical and immunological γHV68-exacerbation of CIA

To examine the requirement of γHV68 latency, as opposed to residual effects from acute infection, for exacerbating CIA, we used a recombinant γHV68 strain that does not develop latency, ACRTA-γHV68. In ACRTA-γHV68, the genes responsible for latency were deleted and a lytic gene, RTA, was constitutively expressed, resulting in clearance of the acute virus by day 14 post-infection (*Rickabaugh et al., 2004*). We infected mice with ACRTA-γHV68, waited 35 days for clearance of the acute infection, and induced CIA. We found that ACRTA-γHV68-infected mice did not develop the CIA clinical enhancement that we observed in latently γHV68-infected mice, with the clinical course and day of onset resembling that of uninfected CIA mice (*Figure 4A–B*).

Furthermore, the immunological changes observed in γHV68-CIA mice, when compared to CIA mice, were absent in ACRTA-γHV68 CIA mice. The increase in relative proportion of CD8+ T cells in

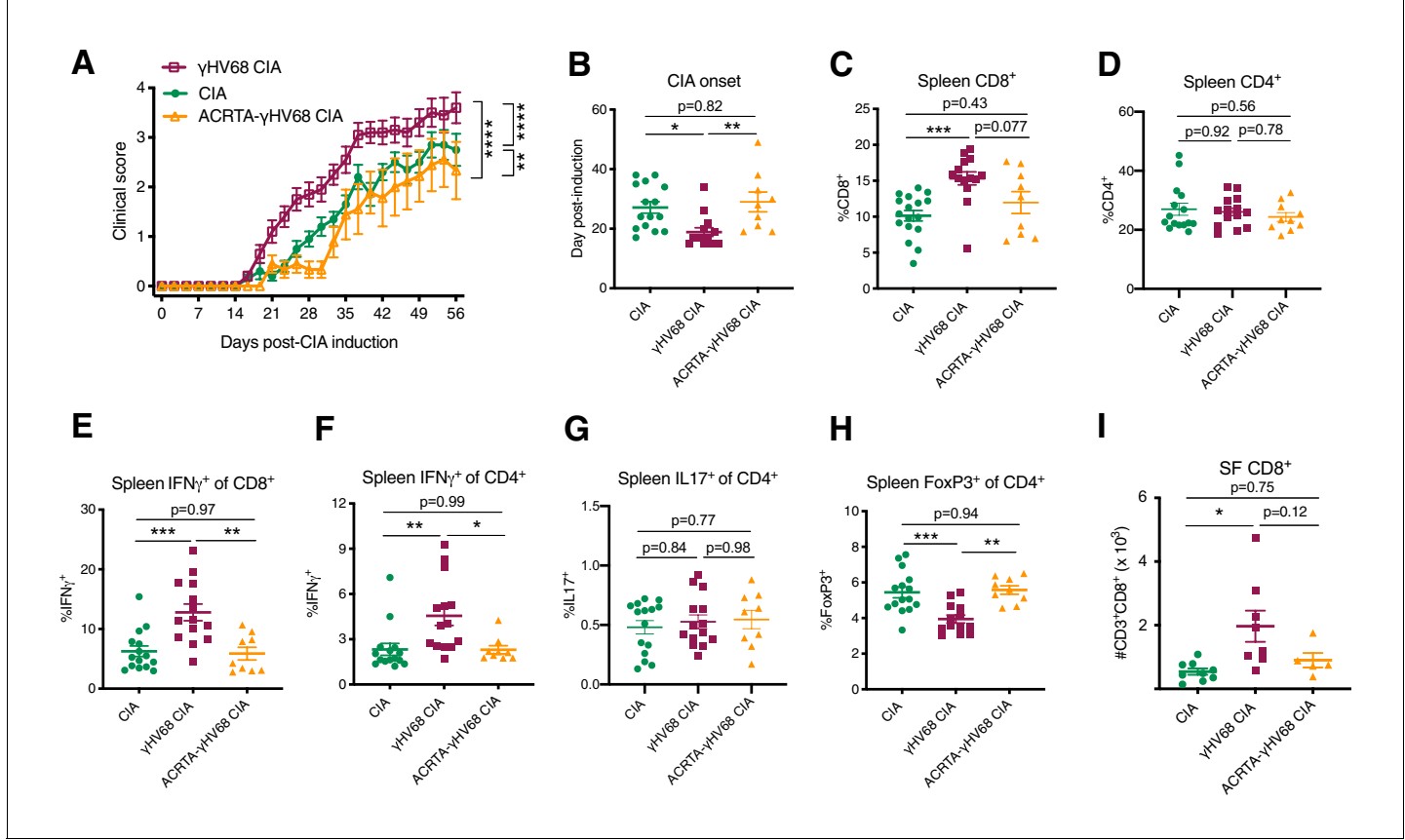

**Figure 4.** Disease progression and immune profile of latency-free ACRTA-γHV68 CIA mice compared to γHV68-CIA and CIA mice. Collagen-induced arthritis (CIA) was induced in C57BL/6J mice after 5 weeks of mock, γHV68, or ACRTA-γHV68 infection, and mice scored for clinical disease until 56 days post-CIA induction. At 56 days post-CIA induction, spleens and synovial fluid were collected and processed for flow cytometry. A proportion of the CIA and γHV68-CIA data is repeated from *Figure 2*. (A) Clinical scores (y-axis) of uninfected mice with CIA (filled circles), γHV68-CIA mice (open squares), and ACRTA-γHV68 CIA mice (open triangles); (B–I) comparison of uninfected mice with CIA (filled circles), γHV68-CIA mice (filled squares), and ACRTA-γHV68 CIA mice (filled triangles). (B) Day (y-axis) of CIA onset, considered two consecutive scoring days of a score of at least 1, in mice (x-axis) without (CIA) and with latent γHV68 infection (γHV68-CIA) or ACRTA-γHV68 infection (ACRTA-γHV68 CIA). (C–I) Immune cell subsets determined by flow cytometry, previously gated on lymphocytes, singlets, live cells, and CD45+ cells; (C) %CD3+CD8+ of CD45+ cells in the spleen; (D) %CD3+CD4+ of CD45+ cells in the spleen; (E) %IFNγ+ of CD8+ cells in the spleen; (F) %IFNγ+ of CD4+ cells in the spleen; (G) IL17A+ of CD4+ cells in the spleen; (H) % FoxP3+ of CD4+ cells in the spleen; (I) number of CD3+CD8+ cells in synovial fluid (SF) determined by flow cytometry. (A–H) n = 9–15 mice per group, two experiments, (I) n = 5–9 mice per group, one experiment. (A–G) Each data point represents an individual mouse. Data presented as mean ± SEM, analyzed by (A) two-way ANOVA with multiple comparisons, F-value = 78.46, (B–I) one-way ANOVA with F-values 6.57 (B), 8.44 (C), 0.053 (D), 11.4 (E), 6.80 (F), 0.29 (G), 10.89 (H), and 5.64 (I); ****p<0.0001, ***p<0.001, **p<0.01, *p<0.05.

The online version of this article includes the following source data for figure 4:

**Source data 1.** Disease progression and immune profile of latency-free ACRTA-γHV68 CIA mice compared toγHV68-CIA and CIA mice.

the spleen was less pronounced in ACRTA-γHV68 CIA compared to γHV68-CIA, while there was no change in relative proportion of CD4$^+$ T cells (*Figure 4C–D*). In ACRTA-γHV68 CIA mice, there was abolishment of the γHV68-induced upregulation of IFNγ in CD8$^+$ and CD4$^+$ T cells, reflecting altered functional capacity and possibly specificity, and no change in IL17A expression by CD4$^+$ T cells (*Figure 4E–G*). The decrease in relative proportion of splenic Tregs and CD8$^+$ infiltration into the synovial fluid observed in γHV68 CIA mice was not present in ACRTA-γHV68 CIA mice (*Figure 4H–I*). Together, these data show that ACRTA-γHV68 CIA mice displayed a similar clinical and immunological profile to uninfected CIA mice. This demonstrates that the enhancement is not due to innate immune stimulation during the acute infection, but, rather, the latency phase of γHV68 infection is critical for the clinical and immunological exacerbation of CIA. The requirement of γHV68 latency mirrors the RA patient clinical course, wherein patients are infected with EBV years before the onset of disease.

## Age-associated B cells are increased and display an inflammatory phenotype in γHV68-CIA

As the number of ABCs was expanded in the contexts of both viral infection and autoimmunity, including RA (*Rubtsov et al., 2011*; *Claes et al., 2016*; *Wang et al., 2018*; *Rubtsova et al., 2013*; *Knox et al., 2017*), we investigated the role of ABCs in facilitating γHV68-exacerbation of CIA. We began by examining the proportion and phenotype of ABCs in uninfected CIA mice and CIA mice previously infected with latent γHV68 (γHV68-CIA) by flow cytometry (*Figure 5—figure supplement 1A*). We found that CIA induction increased the proportion and total number of ABCs (CD19$^+$-CD11c$^+$Tbet$^+$) in the spleen, and γHV68-CIA mice had further increased proportions of ABCs in the spleen compared to CIA (*Figure 5A–B*). The proportion of ABCs in the ILNs was not significantly different between γHV68-CIA and CIA mice (*Figure 5—figure supplement 1C*). The number of ABCs was substantially lower in the ILNs than in the spleen, concurring with other studies that found that ABCs primarily reside in the spleen (*Johnson et al., 2020*). During CIA and γHV68-CIA, we did not observe differences in the proportions of ABCs between male and female mice (*Figure 5—figure supplement 1B*).

We next examined the phenotypic characteristics and found that ABCs in the spleen were phenotypically distinct in γHV68-CIA compared to CIA. We examined a series of markers previously shown to be expressed by ABCs, including cytokines IL10, IFNγ, and TNFα (*Rubtsov et al., 2011*; *Hao et al., 2011*; *Russell Knode et al., 2017*; *Ratliff et al., 2013*), an array of inhibitory receptors (*Rubtsov et al., 2011*; *Wang et al., 2018*; *Knox et al., 2017*), maturity and memory markers IgD, IgM, and CD27 (*Rubtsov et al., 2011*; *Hao et al., 2011*), and MHCII (*Rubtsov et al., 2011*; *Knox et al., 2017*; *Aranburu et al., 2018*). We found that fewer ABCs in the spleens of γHV68-CIA mice expressed IL10, while an increased proportion expressed IFNγ (*Figure 5C*), indicating that they are skewed toward a pathogenic Th1 phenotype. Further, fewer splenic ABCs in γHV68-CIA mice expressed inhibitory receptors CTLA4, PDL1, and PD1 (*Figure 5D–F*), and thus ABCs in CIA displayed a more regulatory phenotype than those in γHV68-CIA mice. Additionally, the ABCs in γHV68-CIA mice displayed a more mature phenotype, with fewer IgD$^+$IgM$^+$ naive B cells and increased MHCII expression, though the expression of memory marker CD27 was unchanged (*Figure 5G–I*). These results indicate that ABCs in γHV68-CIA mice are more mature and may have increased antigen presentation capacities but are not primarily a memory subset. There were no differences in the expression of CD20, TNFα, CD95 (Fas), nor IDO expression (*Figure 5—figure supplement 1D–G*). Collectively, these results indicate that ABCs in γHV68-CIA mice display a more pathogenic phenotype than those in CIA, with decreased expression of regulatory cytokine IL10 and inhibitory markers, and increased expression of IFNγ.

## Age-associated B cells are required for γHV68-exacerbation of CIA

To determine whether ABCs are a subset mediating the viral enhancement of CIA, we utilized ABC KO mice that harbor a B cell-specific Tbet deletion. The clinical course and immune profile of CIA and γHV68-CIA mice were compared in littermate controls of *Tbx21$^{fl/fl}$Cd19$^{cre/+}$* (KO) and *Tbx21$^{fl/fl}$Cd19$^{+/+}$* (Ctrl) mice (*Figure 6A*). We observed that the clinical course was unchanged in CIA between Ctrl and KO mice, indicating that ABCs are not contributing to the disease course in CIA (*Figure 6B*). Alternatively, when induced with CIA, γHV68-infected KO mice did not display the

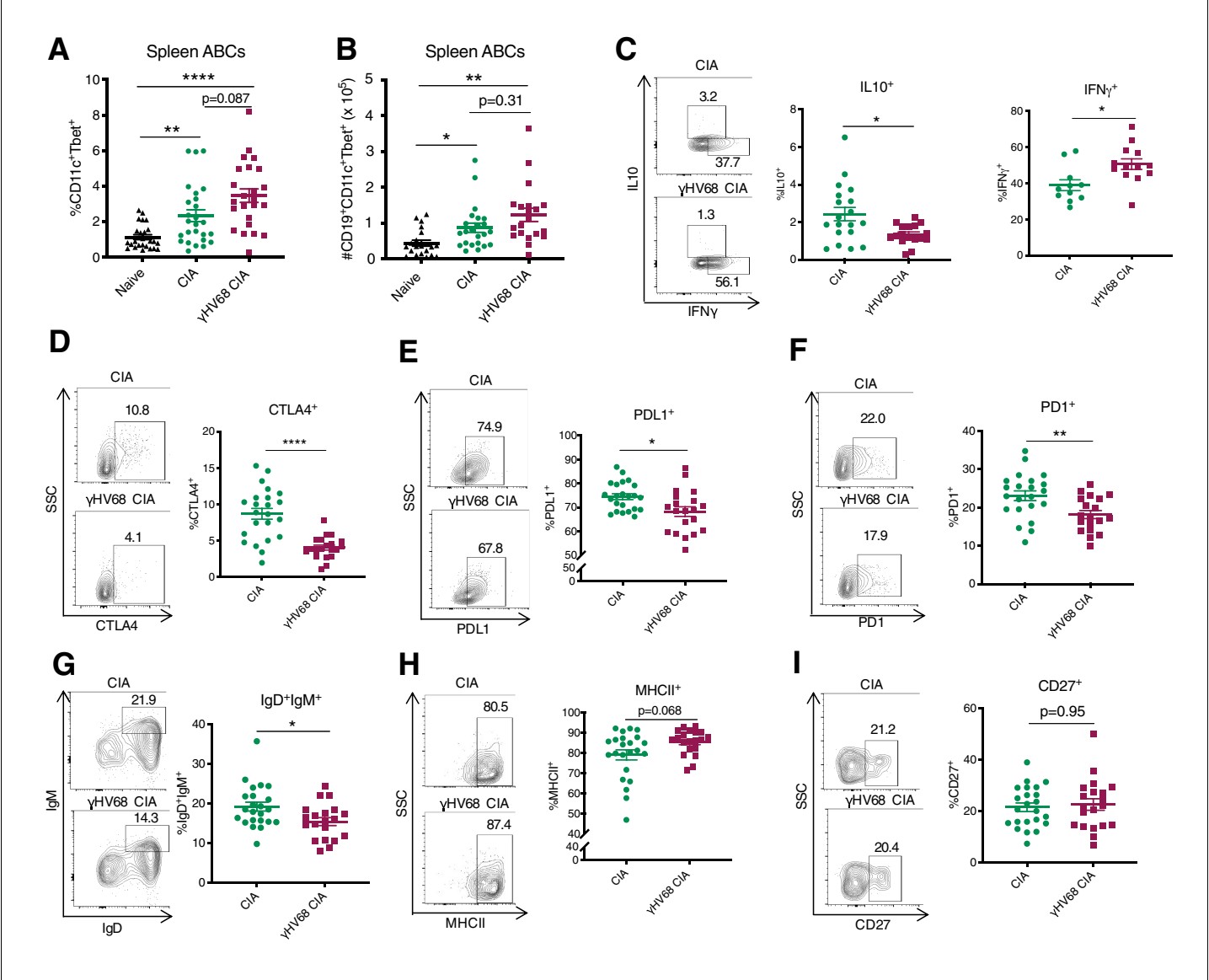

**Figure 5.** Analysis of ABC amount and phenotype by flow cytometry at 56 days post-CIA induction. (**A**) Percentage of age-associated B cells (ABCs) (CD11c⁺Tbet⁺) of mature B cells (CD19⁺IgD⁻) in the spleen and (**B**) total numbers of ABCs in the spleen of naive mice (filled triangle), uninfected mice with collagen-induced arthritis (CIA) (filled circles), and γHV68-CIA mice (filled squares). (**C–I**) Phenotype of ABCs analyzed by flow cytometry. Samples were previously gated on splenic CD19⁺CD11c⁺Tbet⁺ ABCs. Flow plots are representative samples, SSC = side scatter. Proportion of ABCs positive for (**C**) IL10 and IFNγ, (**D**) CTLA4, (**E**) PDL1, (**F**) PD1, (**G**) IgD⁺IgM⁺, (**H**) MHCII, and (**I**) CD27. (**A**) n = 24–26 mice per group, three experiments, (**B**) n = 20–23 mice per group, three experiments, (**C**) n = 16–19 mice per group, two experiments, and (**D–I**) n = 20–23 mice per group, two experiments. (**A–I**) Each data point represents an individual mouse. Data presented as mean ± SEM, (**A–B**) analyzed by Brown-Forsythe and Welch ANOVA tests and (**C–I**) Mann-Whitney test; ****p<0.0001, ***p<0.001, **p<0.01, *p<0.05.

The online version of this article includes the following source data and figure supplement(s) for figure 5:

**Source data 1.** Analysis of ABC amount and phenotype by flow cytometry at 56 days post-CIA induction.

**Figure supplement 1.** ABCs in the spleen analyzed by flow cytometry at 56 days post-CIA induction.

**Figure supplement 1—source data 1.** ABCs in the spleen analyzed by flow cytometry at 56 days post-CIA induction.

γHV68-exacerbated clinical course compared to γHV68-CIA Ctrl mice (*Figure 6C*), indicating that ABCs are a pathogenic subset in γHV68-CIA. Without ABCs, γHV68-CIA mice did not display clinical exacerbation, but rather appeared similar to uninfected CIA mice in terms of disease severity and day of onset (*Figure 6B–D*). We observed that the ablation of ABCs does not significantly alter the proportion of CD8, CD4, or Treg populations in the spleen during CIA or γHV68-CIA, nor the

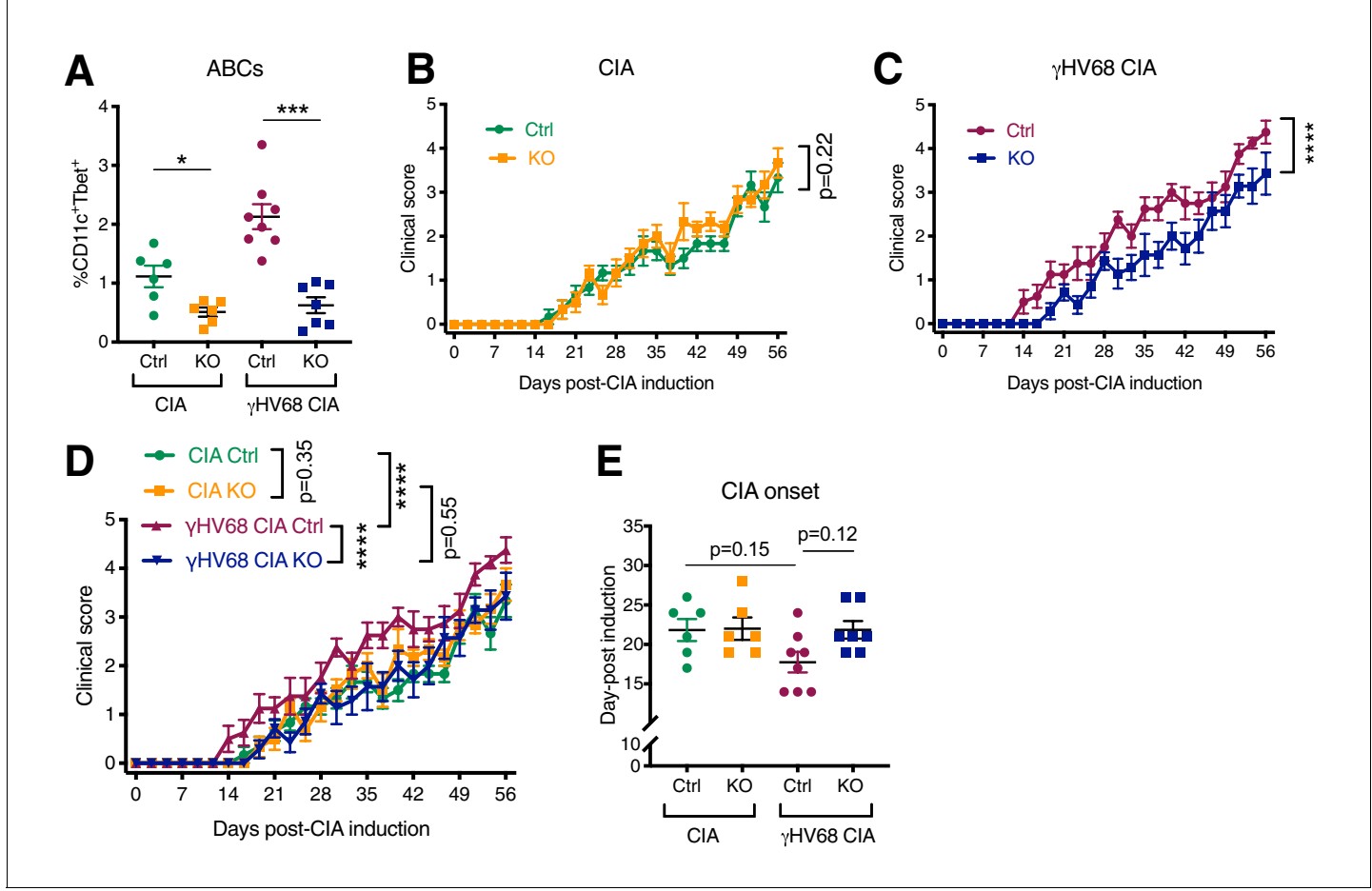

**Figure 6.** Disease progression and flow cytometric analysis of *Tbx21*[fl/fl]*Cd19*[cre/+] (KO) and *Tbx21*[fl/fl]*Cd19*[+/+] (Ctrl) mice that have been infected with γHV68 or mock-infected and induced for CIA. (**A**) Proportion of age-associated B cells (ABCs) (CD11c[+]Tbet[+]) of mature B cells (CD19[+]IgD[-]) in the spleen as determined by flow cytometry of CIA flox-only control (Ctrl, green circles), CIA KO (KO, orange squares), γHV68-CIA flox-only control (Ctrl, purple circles), and γHV68-CIA KO (KO, blue squares) mice. (**B**) Clinical score (y-axis) of collagen-induced arthritis (CIA) measured three times weekly for 8 weeks (x-axis; days) post-CIA induction in Ctrl (green circles) and knockout (KO) (orange circles) mice that are uninfected with CIA. (**C**) Clinical score (y-axis) of CIA measured three times weekly for 8 weeks (x-axis; days) post-CIA induction in Ctrl (purple circles) and KO (blue circles) γHV68-infected CIA mice. (**D**) Clinical scores, same data as in panels (**B, C**). (**E**) Day (y-axis) of CIA onset, considered two consecutive scoring days of a score of at least 1, in Ctrl and KO CIA and γHV68-CIA mice (x-axis). (**A–E**) n = 6–8 mice per group, two experiments. Data presented as mean ± SEM, analyzed by (**A**) Mann-Whitney test, (**B, C**) two-way ANOVA with F-values 1.71 (**B**) and 62.37 (**C**), (**D**) two-way ANOVA with multiple comparisons, with an F-value of 38.14, and (**E**) one-way ANOVA with an F-value of 2.78; ****p<0.0001, ***p<0.001, **p<0.01, *p<0.05.

The online version of this article includes the following source data and figure supplement(s) for figure 6:

**Source data 1.** Disease progression and flow cytometric analysis of *Tbx21*[fl/fl]*Cd19*[cre/+](KO) and *Tbx21*[fl/fl]*Cd19*[+/+](Ctrl) mice that have been infected with γHV68 or mock-infected and induced for CIA.

**Figure supplement 1.** T cell response in the spleen of Ctrl and KO mice at day 56 post-CIA induction.

**Figure supplement 1—source data 1.** T cell response in the spleen of Ctrl and KO mice at day 56 post-CIA induction.

expression of IFNγ or IL17A (*Figure 6—figure supplement 1*). These results indicate that ABCs are a critical pathogenic population in γHV68-CIA though more work is needed to fully elucidate the mechanism by which ABCs are contributing to disease.

## Discussion

In this report, we demonstrate that latent γHV68 exacerbates CIA clinically and immunologically, and Tbet[+] B cells, known as ABCs, are critical for this exacerbation. Investigation of the mechanism by which EBV contributes to RA has previously been challenging due to the lack of a murine model to

examine the systemic immune modulation caused by latent gammaherpesvirus infection and resulting influence on arthritis. Here, we show that infecting mice with latent γHV68 prior to CIA induction results in an immune course more similar to that of RA patients than CIA alone and is a suitable model for examining the contribution of EBV to RA. Elucidating how EBV infection contributes to the development of RA is critical to understanding the underlying pathophysiology of the disease.

As EBV is associated with several autoimmune diseases, it is important to examine whether there are conserved mechanisms of contribution. The overlap in etiology and pathophysiology between these autoimmune diseases may help to explain the cross-efficacy of immunotherapies between MS and RA, including B cell-depletion therapies. Our lab has previously demonstrated that latent γHV68 infection enhances EAE, a common model of MS, clinically and immunologically (*Casiraghi et al., 2012*). In both the γHV68-CIA and γHV68-EAE models, we observed an increase in CD8$^+$ T cells at the site of disease and increased expression of IFNγ by cytotoxic and helper T cells. Latent gammaherpesvirus infection of mice clearly alters autoimmune disease onset and severity reminiscent of the strong association of latent EBV infection in RA patients. As such, these investigative models will serve to identify common mechanisms in which EBV contributes to multiple autoimmune diseases.

Due to EBV infection often taking place years before the onset of arthritis, we posit that latent EBV infection modulates the peripheral immune response in a manner that contributes to the development of RA. We suggest that latently EBV-infected B cells alter, either directly or indirectly, lymphocytes that go on to contribute to disease onset, likely through expanding and activating CD8$^+$ T cells and skewing toward a Th1 response. CD11c$^+$CD8$^+$ DCs may play a role in priming the pathogenic CD8$^+$ T cell response, as they have been shown to cross-present antigen (*den Haan et al., 2000*; *Schulz and Reis e Sousa, 2002*). By acting as a mediator between infected cells and pathogenic T cells, ABCs are likely critical moderators in driving the heightened Th1 immune response to latent viral infection.

Accumulating evidence shows that ABCs are expanded in multiple autoimmune diseases and function pathogenically in mouse models of lupus (*Rubtsov et al., 2011*; *Thorarinsdottir et al., 2019*; *Wang et al., 2019*; *Claes et al., 2016*; *Wang et al., 2018*; *Zhang et al., 2019*; *Rakhmanov et al., 2009*; *Rubtsova et al., 2017*). Precisely how ABCs are contributing to pathogenicity is unclear, and ABCs are known to display multiple functional capacities that could contribute to disease. In models of SLE, ABCs have been shown to secrete autoantibodies and compromise germinal center responses (*Zhang et al., 2019*). Additionally, ABCs function as excellent antigen-presenting cells (*Rubtsov et al., 2015*). In a model of SLE, the ablation of ABCs decreases activated CD4$^+$ T cells and IFNγ-CD8$^+$ T cells (*Rubtsova et al., 2017*). How precisely ABCs alter the CD8$^+$ T cell population, whether they are cross-presenting antigen or impacting the CD8$^+$ T cells indirectly, warrants further investigation. Alternatively, ABCs have been shown to secrete regulatory IL10 (*Rubtsov et al., 2011*; *Hao et al., 2011*), suggesting that a portion of the ABC population, in some individuals or contexts, could function in a protective manner. Further characterization of the phenotype and functional capacities of ABCs in autoimmune patients may help to elucidate their functional role. RA patients who experience a relapse following B cell-depletion therapy are more likely to display a reconstitution profile with increased numbers of memory B cells (*Leandro et al., 2006*). Whether existing therapeutics, such as B cell-depletion therapies or other approved drugs for RA, such as Abatacept (CTLA4 Ig), impact the ABC repertoire remains unknown.

Further evaluation of the influence of viral infection on ABC pathogenicity is needed. It is intriguing that ABCs are pathogenic in a genetic model of SLE without the presence of a virus (*Rubtsova et al., 2017*), though we observe that latent γHV68 is necessary for the pathogenicity of ABCs in CIA. This discrepancy indicates that ABCs may be contributing to disease through various mechanisms or that different contexts can prime ABCs for pathogenicity. The role of ABCs in controlling viral infections is an ongoing topic of study, with multiple papers recently providing compelling evidence that ABCs are critical for an effective anti-influenza response (*Johnson et al., 2020*) and are required to control LCMV infection (*Barnett et al., 2016*), in part through their secretion of antiviral IgG2a. Additionally, the influence of aging on ABC population and on autoimmunity development and progression warrants further study.

In summary, we have developed an in vivo model of EBV's contribution to RA that recapitulates aspects of human disease. Further, we have examined the role of ABCs and found that they are critical mediators of the viral enhancement of arthritis.

# Materials and methods

## Mice

$Tbx21^{fl/fl}Cd19^{cre/+}$ mice were generated by crossing $Tbx21^{fl/fl}Cd19^{cre/+}$ and $Tbx21^{fl/fl}Cd19^{+/+}$ mice. $Tbx21^{fl/fl}$ and $Cd19^{cre/+}$ mice were provided by Dr. Pippa Marrack (*Rubtsova et al., 2017*). C57BL/6J mice were originally purchased from The Jackson Laboratory. All animals were bred and maintained in the animal facility at the University of British Columbia. All animal work was performed per regulations of the Canadian Council for Animal Care (Protocols A17- 0105, A17-0184).

## γHV68 and ACRTA-γHV68 infection

γHV68 (WUMS strain, purchased from ATCC) and ACRTA-γHV68 (originally developed by Dr. Ting-Ting Wu, the generous gift of Dr. Marcia A Blackman) (*Jia et al., 2010*) were propagated in Baby Hamster Kidney (BHK, ATCC) cells. Prior to infection, viruses were diluted in Minimum Essential Media (MEM, Gibco) and maintained on ice. Mice (6- to 8-week-old) were infected intraperitoneally (i.p.) with $10^4$ plaque-forming unit (PFU) of γHV68 or ACRTA-γHV68 or mock-infected with MEM. No clinical symptoms were observed from viral infections.

## Induction of CIA

On day 35 post-γHV68 or -ACRTA-γHV68 infection, CIA was induced by injection of immunization-grade, chick type II collagen emulsified in complete Freund's adjuvant (CFA; Chondrex, Inc) intradermally at the base of the tail, followed by a booster injection of the same emulsion on day 14, as adapted from *Inglis et al., 2008*. Each mouse received 0.1 mg chick type II collagen and 0.25 mg CFA at days 0 and 14.

## Evaluation of CIA severity

Clinical signs of CIA were assessed and scored three times per week beginning at the day of CIA induction: 0 = no symptoms; 1 = slight swelling and/or erythema; 2 = pronounced swelling and erythema; and 3 = severe swelling, erythema, and ankylosis, as adapted from *Brand et al., 2007*. Hind paws were scored individually by a blinded scorer and added for a single score. Day of onset considered two consecutive scoring days of a score of at least 1. The thickness of each hind paw was measured using a digital caliper and the size was expressed as the average thickness of the two paws.

## Tissue harvesting and processing

Mice were anesthetized with isoflurane and euthanized by cardiac puncture. Blood was collected by cardiac puncture into empty sterile tubes and placed on ice until processing, and mice were perfused with 20 ml sterile PBS to allow for synovial fluid harvesting without blood contamination. ILNs and spleen were extracted and placed into 2 ml sterile PBS and stored temporarily on ice until processing. Synovial fluid was collected by exposing the knee and ankle joints, removing the patellar ligament, and flushing each flexed ankle and knee joint with sterile RNase/DNase-free PBS (Invitrogen) using an 18-gauge needle, adapted from *Barton et al., 2007*; *Futami et al., 2012*. Using a 70-μm cell strainer and a 3-ml syringe insert, spleens and ILNs were each mashed through the cell strainer mesh and a single-cell suspension was prepared for each sample. Splenocytes were incubated in ACK lysing buffer for 10 min on ice to lyse red blood cells, and remaining cells were kept on ice until further use.

## Flow cytometry analysis of cell-type-specific surface antigens and intracellular cytokines

To evaluate cytokine production by various cell types, 4 million isolated splenocytes or ILNs were stimulated ex vivo for 3 hr in 5% $CO_2$ at 37°C in Minimum Essential Media (Gibco) containing 10% fetal bovine serum (FBS; Sigma-Aldrich), 1 μl/ml GolgiPlug (BD Biosciences), 10 ng/ml phorbol 12-myristate 13-acetate (PMA, Sigma-Aldrich), and 500 ng/ml ionomycin (Thermo Fisher). Stimulated cells were then washed prior to staining. For each spleen and ILN sample, 4 million cells were stained in two wells, with 2 million cells per well. All collected synovial fluid cells were resuspended in flow cytometry staining buffer (FACS, PBS with 2% newborn calf serum, Sigma-Aldrich) and stained in a single well. Before staining, samples were incubated at 4°C covered from light for 30

min with 2 ul/ml Fixable Viability Dye eFluor506 (Thermo Fisher) while in FACS buffer (PBS with 2% newborn calf serum; Sigma-Aldrich). Cells were then incubated with a rat anti-mouse CD16/32 (Fc block) (BD Biosciences) antibody for 10 min. Fluorochrome-labeled antibodies against cell surface antigens were then added to the cells for 30 min covered from light at 4°C. After washing, cells were suspended in Fix/Perm buffer (Thermo Fisher) for 30 min to 12 hours covered from light at 4°C, washed twice with Perm buffer, and incubated 40 min with antibodies for intracellular antigens in Perm buffer. Cells were then washed and resuspended in FACS buffer with 2 mM ethylenediamine-tetraacetic acid. Cells were stained with anti-mouse CD45 (Clone 30-F11; Thermo Fisher Scientific), CD3 (Clone eBio500A2; Thermo Fisher Scientific), CD19 (Clone eBio1D3; Thermo Fisher Scientific), CD4 (Clone RM4-5; Thermo Fisher Scientific), CD8 (Clone 53–6.7; Thermo Fisher Scientific), FoxP3 (Clone FJK-16S; Thermo Fisher Scientific), IFNγ (Clone XMG1.2; Thermo Fisher Scientific), IL17A (Clone TC11-18H10.1; Thermo Fisher Scientific), IL10 (Clone JES5-16E3; Thermo Fisher Scientific), CD11c (Clone 418; Thermo Fisher Scientific), Tbet (Clone eBio4B10; Thermo Fisher Scientific), CD11b (Clone M1/70; Thermo Fisher Scientific), IgD (Clone 11–26 c; Thermo Fisher Scientific), CTLA4 (Clone UC10-4B9; Thermo Fisher Scientific), PDL1 (Clone MIH5; Thermo Fisher Scientific), PD1 (Clone J43; Thermo Fisher Scientific), IgM (Clone RMM-1; BioLegend), MHCII (Clone M5/114.15.2; BioLegend), CD27 (Clone LG.3A10; BioLegend), CD20 (Clone SA275A11; BioLegend), TNFα (Clone MP6-XT22; BioLegend), CD95 (Clone SA367H8; BioLegend), and IDO (Clone mIDO-48; Thermo Fisher Scientific). The entirety of each sample was collected on an Attune NxT Flow Cytometer (Thermo Fisher) and analyzed with FlowJo software v10 (FlowJo LLC). Full-minus-one (FMO) controls were used for gating.

## *Il17a* and *Ifng* qPCR

RNA was extracted from synovial fluid and ILNs with a Qiagen AllPrep DNA/RNA Micro kit. cDNA was synthesized using Applied Biosystems High-Capacity cDNA Reverse Transcription Kit (Thermo Fisher). qPCR was performed using iQTM SYBR Green supermix (Bio-Rad) on the Bio-Rad CFX96 Touch Real Time PCR Detection system. Primer sets from Integrated DNA Technologies were *Il17a* 5'-GCT CCA GAA GGC CCT CAG-3' (forward) and 5'-AGC TTT CCC TCC GCA TTG-3' (reverse) and *Ifng* 5'-ACT GGC AAA AGG ATG GTG AC-3' (forward) and 5'-TGA GCT CAT TGA ATG CTT GG-3' (reverse). Normalized to the ribosomal housekeeping gene *18* s 5'-GTAACCCGTTGAACCCCATT-3' (forward) and 5'- CCATCCAATCGGTAGTAGCG-3' (reverse) and expression determined relative to control group.

## Anti-type II collagen antibody ELISA

The sera were isolated by centrifugation 2000 × *g* for 10 min, aliquoted, and stored for up to 14 months at −80°C prior to running the enzyme-linked immunosorbent assay (ELISA). Anti-type II collagen antibodies were quantified by standard indirect ELISA. Briefly, ELISA plates (NUNC; Thermo Fisher) were coated with 5 µg/ml ELISA-grade type II collagen (Chondrex, Inc) overnight at 4°C, washed 4x with wash buffer (PBS, 0.05% Tween-20), blocked with 5% newborn calf serum (NBCS; Sigma-Aldrich) for 1 hr at 37°C, incubated with serial dilutions (1:100 to 1:12800) of test sera diluted in blocking buffer for 2 hr at 37°C, and washed 4x with wash buffer. Bound (anti-collagen II) antibody was incubated with HRP-conjugated goat anti-mouse IgG (Thermo Fisher), rat anti-mouse IgG1 (BD Biosciences), or goat anti-mouse IgG2c (Thermo Fisher), all diluted 1:500 in blocking buffer, for 1 hr at 37°C, washed 4x with wash buffer, and detected by TMB substrate (BD Biosciences). Absorbance was read at 450 nm on a VarioSkan Plate Reader (Thermo Fisher).

## γHV68 quantitation

Quantification of γHV68 load was done as previously described (*Márquez et al., 2020*). Genomic DNA (gDNA) was extracted from $4 \times 10^6$ splenocytes with PureLink Genomic DNA mini kit (Thermo Fisher), according to the manufacturer's instructions, and stored at −20°C. For qPCR, 150 ng DNA per reaction was amplified in duplicate using primers and probes specific to γHV68 ORF50 and mouse PTGER2 with QuantiNova Probe Mastermix (Qiagen). Primers and probes used from Integrated DNA Technologies were PTGER2: forward primer: **5'-TACCTTCAGCTGTACGCCAC-3'**; reverse primer: **5'-GCCAGGAGAATGAGGTGGTC-3'**; probe: **5'-/56-FAM/CCTGCTGCT/ZEN/TATCGTGGCTG/3IABkFQ/-3'**; ORF50: forward primer: **5'-TGGACTTTGACAGCCCAGTA-3'**; reverse

primer: **5′-TCCCTTGAGGCAAATGATTC-3′**; probe: **5′-/56-FAM/TGACAGTGC/ZEN/CTA TGGCCAAGTCTTG/3IABkFQ/-3′**. Standard curves were obtained by serial dilutions of ORF50 and PTGER2 gBlocks (ORF50: $2 \times 10^6 - 2 \times 10^1$; PTGER2: $5 \times 10^7 - 5 \times 10^2$). Reactions were run on the Bio-Rad CFX96 Touch Real Time PCR Detection system.

## Statistics

Data and statistical analyses were performed using GraphPad Prism software 8.4.2 (GraphPad Software Inc). Results are presented as mean ± SEM. Statistical tests, significance (p-value), sample size (n, number of mice per group), and number of experimental replicates are stated in figure legends. Statistical analyses included two-way ANOVA with Geisser-Greenhouse's correction, Mann-Whitney test, or one-way ANOVA. P-values are indicated by asterisks as follows: ****$p<0.0001$, ***$p<0.001$, **$p<0.01$, *$p<0.05$.

## Study approval

All work was approved by the Animal Care Committee (ACC) of the University of British Columbia (Protocols A17- 0105, A17-0184).

## Acknowledgements

We are grateful to Dr. Philippa Marrack for providing *Tbx21*<sup>fl/fl</sup> and *Cd19*<sup>cre/+</sup> mice and to Jessica R Allanach for advice and help in preparing the manuscript. This research was supported in part by the MSSC, CIHR, UBC, and JDRF.

## Additional information

### Funding

| Funder | Grant reference number | Author |
|---|---|---|
| Multiple Sclerosis Society of Canada | # 3070 | Marc S Horwitz |

The funders had no role in study design, data collection and interpretation, or the decision to submit the work for publication.

### Author contributions

Isobel C Mouat, Conceptualization, Data curation, Formal analysis, Validation, Investigation, Methodology, Writing - original draft, Writing - review and editing; Zachary J Morse, Iryna Shanina, Investigation, Methodology; Kelly L Brown, Writing - review and editing; Marc S Horwitz, Conceptualization, Resources, Supervision, Funding acquisition, Writing - original draft, Project administration, Writing - review and editing

### Author ORCIDs

Isobel C Mouat https://orcid.org/0000-0003-4319-9067
Marc S Horwitz https://orcid.org/0000-0002-5922-2199

### Ethics

Animal experimentation: All work was approved by the Animal Care Committee (ACC) of the University of British 504 Columbia (Protocols A17- 0105, A17-0184).

### Decision letter and Author response

Decision letter https://doi.org/10.7554/eLife.67024.sa1
Author response https://doi.org/10.7554/eLife.67024.sa2

## Additional files

### Supplementary files
• Transparent reporting form

### Data availability
All data generated or analysed during this study are included in the manuscript and supporting files.

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
