## [Decision Letter]

**Acceptance summary:**

This is a very interesting study and could help shed light on mechanistic connections between latent infection by EBV with an age-dependent autoimmune condition, such as rheumatoid arthritis (RA).The authors use two models: a murine model of rheumatoid arthritis (Collagen-Induced Arthritis, CIA), and a murine analog of human EBV: γHV68. The use of these two models allows the investigation of how latent viral infection exacerbates the autoimmune condition via the action of a special class of B cells: Age-associated B cells.

**Decision letter after peer review:**

Thank you for submitting your article "Latent gammaherpesvirus exacerbates arthritis and requires age-associated B cells" for consideration by *eLife*. Your article has been reviewed by 3 peer reviewers, and the evaluation has been overseen by a Reviewing Editor and Betty Diamond as the Senior Editor. The following individual involved in review of your submission has agreed to reveal their identity: Lena Pernas (Reviewer #2).

Essential revisions:

This is a very interesting study could help shed light on mechanistic connections between latent infection by EBV with an age-dependent autoimmune condition, such as rheumatoid arthritis. The authors use two models: a murine model of rheumatoid arthritis (CIA), and a murine analog of human EBV: γHV68. The use of these two models allows the investigation of how latent viral infection exacerbates the autoimmune condition via the action of a special class of B cells: Age-associated B cells.

While all three reviewers recognise the potential of this work, an essential revision is needed before this work can be considered acceptable for publication in *eLife*. I would like to list the essential reviews needed, and I'd like to invite the authors to address all the comments from the reviewers.

1. Statistics: please address all questions regarding statistical analyses raised by reviewer #1.

2. The authors need to explain upfront why they chose to use young mice for their study, instead of aged subjects.

3. Is the acute collagen injection reactivating γHV68? The authors assume that the immune responses are largely due to latent viral infection. However, can they rule out a virus reactivation due to the experimental model of CIA?

4. The title needs to be revised to more faithfully reflect the results presented – in particular, the role of ABC is not clear from the current title.

5. Most of the comparisons made (as presented in figure 4) compare the ACRTA-γHV68 CIA mice to γHV68 CIA mice – my question is – does the infection with the mutated γHV68 affect these parameters in comparison to CIA mice?

*Reviewer #1 (Recommendations for the authors):*

Following are a few comments that concern clarity and rigor of the data analysis and manuscript.

1. There are a number of concerns regarding the completeness and appropriateness of a number of statistical analyses.

a. Authors tend to use t-tests throughout the manuscript. T-tests are not appropriate if the data is not normally distributed. In general, a test for normality should be performed (e.g. Shapiro) and mentioned in legends and methods if t-test must be used. Alternatively, it is usually better to choose non-parametric tests (e.g. Wilcoxon or Mann-Whitney) which do not have such prerequisites for usage. Thus, the statistics should be reviewed carefully in any future submission.

b. Two-way ANOVA analysis does not seem to be appropriate for longitudinal data, such as in Figure 1A. Data analysis methods more suitable for time-lapse datasets, such as area under curve, should be used. In general, a more detailed description of statistical analyses and results used in the manuscript should be added – eg. F-values for ANOVA, etc.

c. In addition to a "time-course" analysis, end-point statistical test results should be provided for clinical score data – eg. Figure 1A.

d. Significance test results between CIA and ACRTA-γHV68 CIA mice should also be included in Figures 4B-I.

e. Significance of difference in clinical scores between CIA and γHV68-CIA in Figure 4A is significantly lower compared to that from Figure 1A. Can the authors comment on this?

2. The authors use very young mice (6-8 weeks) to study an age-related condition, RA. At that age, mice are barely sexually mature. This a a potential caveat/confound of the study and should be extensively discussed in the manuscript (for instance, discussing how studying the impact of latency in >15 months old mice may alter conclusions).

3. Data from Figure 4A and Figure 4C do not seem to correlate: ACRTA-γHV68 CIA mice present with lower clinical scores compared to γHV68 CIA mice in Figure 4A, but spleen CD8^+^ levels are elevated in ACRTA-γHV68 CIA mice relative to γHV68 CIA mice. Additionally, spleen CD8^+^ proportions show greater variance in ACRTA-γHV68 CIA mice. Authors should discuss these points more extensively in the manuscript.

4. When comparing clinical score data from ABC Ctrl and KO mice, datasets from CIA and γHV68-CIA mice (e.g. Figure 6B and C) should be plotted together instead (or at least in an additional panel if needed). When needed, please note statistical tests should be corrected for multiple comparisons.

5. In the manuscript, authors state that no difference was observed between males and females in terms of effects of γHV68 on CIA progression. However, since it is well-known that women are significantly more likely to develop RA compared to men, authors should discuss how the findings from this study should be applied to future clinical studies. Additionally, in clinical score data from Supplementary Figure 1B, male mice seem to present with greater difference in clinical scores between CIA and 𝜸HV68-CIA mice compared to females. Comments on these observations should be added to the manuscript.

*Reviewer #2 (Recommendations for the authors):*

Regarding the role of 'latent' gammaherpesvirus:

1) The authors conclude that latent rather than acute EBV is responsible for exacerbating the symptoms of CIA and systematically modulating immune traits. However, the authors never assess whether the stress of collagen injection reactivates γHV68 (i.e. is there a spike in anti- γHV68 IgG?). The authors should check this or restate the conclusions regarding 'latent gammaherpesvirus' infection.

2) Along the previous comment, it is further not entirely clear how the authors executed their experiments with the acute strain ACRTA. The current methods are not detailed enough and should precisely state when the acute infection was performed/when infection is cleared. To demonstrate that acute infection is truly not modulating immune parameters or exacerbating disease symptoms, acute infection should have been performed at day 35 (depending on how long it takes to clear the pathogen) – is this the case?

Further clarification of immunological experimentation:

3) The authors should clarify why they analyzed certain cytokines, as an analysis of a broader set of cytokines would have been clearly valuable given our limited understanding of the role of chronic infections in shaping immune parameters.

4) The authors are overstating some of their findings given several non-significant reports (see comments below).

Title: needs be rephrased as it currently reads as if EBV benefits from the presence of ABCs.

Abstract:

Line 31/32: it remains unknown whether ABCs are directly stimulated by the virus infection or whether the presence of e.g. certain cytokines/immune cells due to infection are stimulating the appearance of ABCs. Thus "infection stimulates ABCs" should be rephrased as the authors do not provide evidence for a direct causation in their work.

Introduction:

Line 42-44: previous work stated in this line reports that RA patients have higher loads of EBV, indicating acute EBV in RA patients. Is anything known how often EBV shifts between lytic and latent stage in these patients? And how does it relate to what the authors conclude in their current work?

Line 47-50: for a better reading flow this should be moved after the sentence in line 42.

Line 52-55: references should be added

Line 55: was this examined during an active or chronic EBV infection?

Line 64/65: this sentence is vague – perhaps 'enhancement' could be clarified

Line 69: does this strain reactive regularly? If yes, how did authors ensure in their experiments that the used EBV strain was not lytic during the onset of symptoms or tissue collection 56 days post-latency?

Line 80: typo in C57BI/6 mice.

Line 83: given that the author's experiments do not truly uncover a mechanism by which EBV modulates RA disease, this sentence should be rephrased.

Line 86: typo – to the contribution of EBV

Line 93-95: references should be provided

Results:

Line 110: point scale is later on reported to range from 0-3.

Line 115/116: reference(s) should be added.

Line 155: more detailed explanation why specifically IFNγ and IL-17 two cytokines were chosen. See e.g. line 211.

Line 156: typo in fold-change 129

Line 157: authors report a trend towards a reduced relative expression of IL-17, while the p-value is clearly non-significant (p=0.24). Same for line 189 with a p-value of p=0.19. Accordingly, findings should be rephrased.

Line 242: how does this relate to findings from reference 3 that RA patients have higher EBV loads, indicating that it is in a lytic stage?

Line 267: not clear what is meant with viral enhancement?

Line 270: giving the non-significant p-values shown in Figure 5 B (p=0.077 and p=0.16), the total number of ABCs is not increased. Accordingly, findings and conclusions should be carefully rewritten.

Line 283/284: this sentence is vague.

Line 285-287: authors need to give more insights why certain parameters were examined.

Line 317: related Figures should be C and D, not D and E.

Line 317/318: results are overstated given the weak statistical support (p=0.07).

Line 320/321: can this be added to the supplements?

Figures:

All p-values should be reported consistently (e.g. Figure 2A "ns", Figure 2B and C "p=0.24" and "p=0.22"). See also Figure 4.

Stats are missing in Figure 3 D "ILN IFNγ of CD4" and Figure 3E, 6F

Figure 6H (line 320) is missing completely

Only figure 6 is in color, while the rest is black/white

Material and Methods:

Line 416: the authors should provide further details on the used strains, e.g. can/does yHV68 reactivate? When is the lytic strain ACRTA cleared?

*Reviewer #3 (Recommendations for the authors):*

The manuscript entitled: "Latent gammaherpesvirus exacerbates arthritis and requires age-associated B cells" by Mouat et al. addresses the question of whether gammaherpesvirus in its latent form, contributes to the severity of clinical RA and if latent gammaherpesvirus infection enhances clinical arthritis. To address this question, they used a mouse model where they monitored the severity of collagen-induced arthritis (CIA). They demonstrate that disease enhancement requires viral latency and that this enhancement is not due to active virus stimulation of the immune response. They investigated the association between age-associated B cells and the progression of arthritis. Using ABC knockout mice, they provided the first evidence that ABCs are mechanistically required for viral enhancement of disease, thereby establishing that latent gammaherpesvirus infection stimulates ABCs to provoke arthritis.

They used the C57Bl/6 mice model and infected them with EBV homolog, gHV68, waited 5 weeks for the lytic infection to clear thus, modeling the latent form of the infection. The induction of latency was used in three different scenarios: i) gHV68 only (latent); ii) collagen-induced arthritis (CIA) and; iii) the combination of gHV68 latent infection with CIA. They monitored the severity of the disease by redness and swelling in the hind two paws to inform a clinical score (scale of 0-4) and the δ of paw size. They observe that CIA in latent gHV68-infected mice has a more severe clinical course than uninfected mice and developed the onset of disease symptoms an average of seven days earlier than CIA-only mice. They also found that latent gHV68 infection did not affect autoantibody levels thus, latent gHV68 infection leads to the earlier onset and more severe CIA.

Next, the authors assessed the frequency of immune cells that infiltrates to the synovium (fluid in a specialized connective tissue) by flushing each joint with a buffer (PBS) and analyzing the composition using flow cytometry. The focus was on CD8^+^ and CD4^+^ T cell. They found that gHV68-CIA comprises 3.6-fold more CD8^+^ T cell in gHV68-CIA mice compared to CIA only. They did not find any change in the proportion of CD4^+^ T cells. Moreover, they found a significant increase of Tbet (Th1 cell-specific transcription factor) in both CD4^+^ and CD8^+^ that is indicative to Th1 skewing and further tested the levels of IFNγ and IL-17.

They state that the results indicate that IFNγ-producing T cells are preferentially infiltrating the synovium of gHV68-CIA mice, and a skewing towards cytotoxic CD8^+^ T cells in mice latently infected with gHV68 prior to CIA.

To assess if latent gHV68 infection skews the T cell response towards a pathogenic profile during CIA, the authors examined T cells in the spleen, inguinal/draining lymph nodes and found that gHV68-CIA mice display a decrease in the relative proportion of FoxP3+ Tregs and an increase in the relative proportion of CD8^+^ T cells in the spleen compared to control CIA mice. They show overall that the T cell profile of gHV68-CIA mice is skewed pathogenically, with decreased Tregs and increased cytotoxic T cells.

To demonstrates that infection with latent gHV68 is responsible for the enhancement of CIA and this is not due to innate immune stimulation during the acute infection the authors describe the response following the infection with a recombinant gHV68 strain that does not develop latency, namely, ACRTA-gHV68. They found that the immunological changes observed in gHV68-CIA mice, when compared to CIA mice, are absent in ACRTA-gHV68 CIA mice.

To investigate the role of age-associated B cells (ABCs) to facilitate the viral enhancement of CIA they studied the proportion and phenotype of ABCs in uninfected CIA mice and CIA mice previously infected with latent gHV68 (gHV68-CIA) by flow cytometry. These experiments indicate that ABCs in gHV68-CIA mice display a more pathogenic phenotype than those in CIA, with decreased expression of regulatory cytokine IL10 and inhibitory markers, and increased expression of IFNγ.

Moreover, they utilized ABC knock-out mice that harbor a B cell-specific Tbet deletion. Without ABCs, gHV68-CIA mice do not display clinical exacerbation, but rather appear similar to uninfected CIA mice in terms of disease severity and day of onset.

Overall, the authors show that:

– Infecting mice with latent gHV68 prior to CIA induction results in an immune course more similar to that of RA patients than CIA alone, and is a suitable model for examining the contribution of EBV to RA.

– Tbet+ B cells, known as age-associated B cells, are critical for this CIA exacerbation

The author suggests that:

– Latently EBV-infected B cells alter, either directly or indirectly, lymphocytes that go on to contribute to disease onset, likely through expanding and activating CD8^+^ T cells and skewing towards a Th1 response.

The manuscript is written well and the read flows. The experiments and results support the conclusions and this report is publishable.

I have several comments/revisions/corrections and questions regarding the data:

– Line 70: missing space between the letter "a" and the abbreviation "gHV68"

– Line 80: typo is the mouse strain – written "C7Bl/6" and should be "C57Bl/6"

– Line 86:- "The role of B cells the contribution of EBV to RA is intriguing…", correct the grammar

– Line 94: In line 63 the abbreviation "lymphocytic choriomeningitis virus (LCMV)" was defined so when mentioning it the second time it should state the abbreviation and not the full name.

– Line 146: The authors state that "Synovial cells were not collected from naïve or gHV68-infected mice without CIA because we would not expect there to be sufficient infiltration of immune cells for analysis". On what basis do the author state this? Was it established that immune cells do not infiltrate the synovium w/o CIA?

– The author state (line 238) "… that ACRTA-gHV68 CIA mice display a similar clinical and immunological profile to uninfected CIA mice". However, in figure 4 they compare the ACRTA-gHV68 CIA group to gHV68 CIA and not to CIA only group. It would be important to see that there is no significant difference between the CIA and ACRTA-gHV68 CIA group to support their statement.

– Line 156 – fold chage -> fold change

– Line 157: The author describes an increase in IFNγ and decrease in IL-17 however, there is no statistical significance using the Mann-Whitney t-test. Although they describe a fold increase, they do not comment on the high p values.

– Line 164: Figure 2A – it is not clear how the numbers in the figure were calculated – % out of CD45+CD3+ cells? Moreover, not clear to what does the description "…filled circles) and γHV68-CIA mice (filled squares)" relates to in the figure legend?

– Line 188: The increase of CD8^+^ and decrease of FoXP3+ in the spleen of gHV68-CIA mice vs CIA mice is compared to a similar observation in the circulation of humans. Does this mean that the education process of T cells in the spleen is mirrored to the circulation? it would be interesting to check the same in the mouse circulation? (it is possible to recover PBMCs from mice )

– Line 189: " In the ILNs of gHV68-CIA mice, we observe a trend of the deceased relative proportion of regulatory T cells and…" – from figure 3A it does not seem that there is a decrease in Treg in the ILN of gHV68-CIA mice. Could the authors clarify this point?

– Line 228: When the authors write resembling uninfected CIA mice – did they check the significance of CIA onset (figure 4B) between CIA and ACRTA-γHV68 CIA ?

– Most of the comparisons made (as presented in figure 4) compare the ACRTA-γHV68 CIA mice to γHV68 CIA mice – my question is – does the infection with the mutated γHV68 affect these parameters in comparison to CIA mice?

---

## [Author Response]

Essential revisions:This is a very interesting study could help shed light on mechanistic connections between latent infection by EBV with an age-dependent autoimmune condition, such as rheumatoid arthritis. The authors use two models: a murine model of rheumatoid arthritis (CIA), and a murine analog of human EBV: γHV68. The use of these two models allows the investigation of how latent viral infection exacerbates the autoimmune condition via the action of a special class of B cells: Age-associated B cells.While all three reviewers recognise the potential of this work, an essential revision is needed before this work can be considered acceptable for publication in eLife. I would like to list the essential reviews needed, and I'd like to invite the authors to address all the comments from the reviewers.1. Statistics: please address all questions regarding statistical analyses raised by reviewer #1.

We appreciate the comments from reviewer #1 regarding statistical analyses. We have updated all of the statistical analyses in the manuscript as per their comments.

2. The authors need to explain upfront why they chose to use young mice for their study, instead of aged subjects.

It is always difficult to make precise age comparisons between different species, especially mice and humans, but in this study, we have attempted to recapitulate the timing of infection and disease. We have infected mice with γHV68 when they are young but still sexually and immunologically mature (6-8 weeks-old) and then have induced CIA 5 weeks later, when mice are 11-13 weeks old and virus has well established latency. This mirrors the human condition as EBV infection typically occurs in childhood or late adolescence and RA is observed later during adulthood.

3. Is the acute collagen injection reactivating γHV68? The authors assume that the immune responses are largely due to latent viral infection. However, can they rule out a virus reactivation due to the experimental model of CIA?

Our previous work shows that adjuvant (CFA), that is used in both EAE and CIA induction, does not result in γHV68 reactivation (Casiraghi, 2012). We now provide qPCR data that demonstrates that the viral load is not elevated in γHV68-CIA mice (Figure 1—figure supplement 1I).

4. The title needs to be revised to more faithfully reflect the results presented – in particular, the role of ABC is not clear from the current title.

The title now reads as “Latent gammaherpesvirus exacerbates arthritis through modification of age-associated B cells”

5. Most of the comparisons made (as presented in figure 4) compare the ACRTA-γHV68 CIA mice to γHV68 CIA mice – my question is – does the infection with the mutated γHV68 affect these parameters in comparison to CIA mice?

We have added comparisons between CIA and ACRTA-γHV68 CIA and find no difference between these groups, showing that only γHV68 that develops latency drives the exacerbation of disease.

Reviewer #1 (Recommendations for the authors):Following are a few comments that concern clarity and rigor of the data analysis and manuscript.1. There are a number of concerns regarding the completeness and appropriateness of a number of statistical analyses.a. Authors tend to use t-tests throughout the manuscript. T-tests are not appropriate if the data is not normally distributed. In general, a test for normality should be performed (e.g. Shapiro) and mentioned in legends and methods if t-test must be used. Alternatively, it is usually better to choose non-parametric tests (e.g. Wilcoxon or Mann-Whitney) which do not have such prerequisites for usage. Thus, the statistics should be reviewed carefully in any future submission.

Thank you for pointing this out. We have changed t-tests to Mann-Whitney tests throughout the manuscript.

b. Two-way ANOVA analysis does not seem to be appropriate for longitudinal data, such as in Figure 1A. Data analysis methods more suitable for time-lapse datasets, such as area under curve, should be used. In general, a more detailed description of statistical analyses and results used in the manuscript should be added – eg. F-values for ANOVA, etc.

Thank you for these helpful suggestions. We have added area under the curve values for each individual mouse to supplementary figure 1. We have also re-run ANOVAs to account for multiple comparisons and have added F-values to the figure descriptions for all ANOVA tests throughout the manuscript.

c. In addition to a "time-course" analysis, end-point statistical test results should be provided for clinical score data – eg. Figure 1A.

Thank you for this suggestion, we have added both endpoint and cumulative clinical scores to the supplement.

d. Significance test results between CIA and ACRTA-γHV68 CIA mice should also be included in Figures 4B-I.

Thank you for pointing this out, we have added this comparison.

e. Significance of difference in clinical scores between CIA and γHV68-CIA in Figure 4A is significantly lower compared to that from Figure 1A. Can the authors comment on this?

Thank you for pointing this out. We have revised the statistical tests to account for multiple comparisons and now find that the differences to be the same between Figure 4A and Figure 1A.

2. The authors use very young mice (6-8 weeks) to study an age-related condition, RA. At that age, mice are barely sexually mature. This a a potential caveat/confound of the study and should be extensively discussed in the manuscript (for instance, discussing how studying the impact of latency in >15 months old mice may alter conclusions).

Thank you for your concern. The role of age and time between infection and disease onset is something we discuss extensively in our group. In modeling disease associated with EBV, we need to take into consideration that over 95% of EBV infections take place in childhood, adolescence, and early adulthood (before 20 years of age). Prevalence of RA increases with age, typically occurs after the age of EBV infection during middle-age, roughly between 30-60. It is always difficult to make precise age comparisons between different species, especially mice and humans, but general age comparisons can be made. For mice, adolescence and early adulthood is modelled in immunologically and sexually mature mice between 6-8 weeks of age. We make it a practice to only use the fully developed mice. CIA induction takes place 5 weeks post-infection when the mice are well into adulthood. So these are not “very young mice.” We have added this sentence to the introduction to further clarify: “EBV primary infection generally takes place in childhood or adolescence(6,7) and RA can occur at any age, though the mean incidence occurs during middle-age(23). Accordingly, we have infected immunologically and sexually mature 6-8-week-old C57BL/6J mice with γHV68 and have induced CIA when the mice are adults at 11-13 weeks old.”

3. Data from Figure 4A and Figure 4C do not seem to correlate: ACRTA-γHV68 CIA mice present with lower clinical scores compared to γHV68 CIA mice in Figure 4A, but spleen CD8^+^ levels are elevated in ACRTA-γHV68 CIA mice relative to γHV68 CIA mice. Additionally, spleen CD8^+^ proportions show greater variance in ACRTA-γHV68 CIA mice. Authors should discuss these points more extensively in the manuscript.

We appreciate this observation. While there may be higher proportions of CD8 T cells in ACRTA-γHV68, they express significantly less IFNγ (Figure 4E), reflecting altered functional capacity and possibly specificity. We have added this to the text (line 298).

4. When comparing clinical score data from ABC Ctrl and KO mice, datasets from CIA and γHV68-CIA mice (e.g. Figure 6B and C) should be plotted together instead (or at least in an additional panel if needed). When needed, please note statistical tests should be corrected for multiple comparisons.

Thank you for pointing this out, we have added an additional panel that displays all datasets together (Figure 6D).

5. In the manuscript, authors state that no difference was observed between males and females in terms of effects of γHV68 on CIA progression. However, since it is well-known that women are significantly more likely to develop RA compared to men, authors should discuss how the findings from this study should be applied to future clinical studies. Additionally, in clinical score data from Supplementary Figure 1B, male mice seem to present with greater difference in clinical scores between CIA and 𝜸HV68-CIA mice compared to females. Comments on these observations should be added to the manuscript.

While prevalence of RA occurs to a greater extent in women, in CIA induction in mice there is no sex prevalence. As an induced model, it likely breaks through some of the sex related susceptibility/resistance. Similarly, CIA has little association to age. We highlight in the text that this is in line with previous findings: “In agreement with other research groups(22), male and female mice displayed similar clinical scores during CIA and we also do not observe a sex difference in γHV68-CIA mice (Figure 1—figure supplement 1E)” (line 146).

Reviewer #2 (Recommendations for the authors):Regarding the role of 'latent' gammaherpesvirus:1) The authors conclude that latent rather than acute EBV is responsible for exacerbating the symptoms of CIA and systematically modulating immune traits. However, the authors never assess whether the stress of collagen injection reactivates γHV68 (i.e. is there a spike in anti- γHV68 IgG?). The authors should check this or restate the conclusions regarding 'latent gammaherpesvirus' infection.

Thank you for this suggestion. We now provide qPCR data that demonstrates that the viral load is not elevated in γHV68-CIA mice (Figure 1 —figure supplement 1I). These results and methods have been added to the supplementary materials and we have added these sentences to the text: “Additionally, we find that inducing CIA in γHV68-infected mice does not impact viral load (Figure 1—figure supplement 1I), indicating that γHV68 is not reactivating. These findings are in line with our previous work showing that latent γHV68 infection enhances EAE without influencing autoantibody levels or reactivating γHV68(18)” (line 159).

2) Along the previous comment, it is further not entirely clear how the authors executed their experiments with the acute strain ACRTA. The current methods are not detailed enough and should precisely state when the acute infection was performed/when infection is cleared. To demonstrate that acute infection is truly not modulating immune parameters or exacerbating disease symptoms, acute infection should have been performed at day 35 (depending on how long it takes to clear the pathogen) – is this the case?

Thank you, we apologize for this oversight. We have added further details to the Results section to clarify, which now reads: “In ACRTA-γHV68 the genes responsible for latency are deleted and a lytic gene, RTA, is constitutively expressed, resulting in clearance of the acute virus by day 14 post-infection(48). We infected mice with ACRTA-γHV68, waited 35 days for clearance of the acute infection, and induced CIA” (line 287).

To the methods we have added: “On day 35 post-γHV68 or ACRTA-γHV68 infection, CIA was induced by injection of immunization-grade, chick type II collagen emulsified in complete Freund’s adjuvant (CFA, Chondrex, Inc) intradermally at the base of the tail, followed by a booster injection of the same emulsion on day 14, as adapted from(59)” (line 538)

Further clarification of immunological experimentation:3) The authors should clarify why they analyzed certain cytokines, as an analysis of a broader set of cytokines would have been clearly valuable given our limited understanding of the role of chronic infections in shaping immune parameters.

We agree that examining a wider array of cytokines would be valuable. Our initial approach was to focus our examination on immediately relevant cytokines from our previous work where we show altered chronic disease during latent γHV68 infection. We are currently following up on these findings with a broader examination of the cytokine expression profile.

4) The authors are overstating some of their findings given several non-significant reports (see comments below).Title: needs be rephrased as it currently reads as if EBV benefits from the presence of ABCs

Thank you for pointing this out. The title now reads as “Latent gammaherpesvirus exacerbates arthritis through modification of age-associated B cells”.

Abstract:Line 31/32: it remains unknown whether ABCs are directly stimulated by the virus infection or whether the presence of e.g. certain cytokines/immune cells due to infection are stimulating the appearance of ABCs. Thus "infection stimulates ABCs" should be rephrased as the authors do not provide evidence for a direct causation in their work.

Thank you for pointing this out, this sentence has been rephrased (line 33).

Introduction:Line 42-44: previous work stated in this line reports that RA patients have higher loads of EBV, indicating acute EBV in RA patients. Is anything known how often EBV shifts between lytic and latent stage in these patients? And how does it relate to what the authors conclude in their current work?

In Balandraud et al., the researchers measured EBV DNA and this does not indicate replicating or lytic virus, just overall more virus. The reactivation dynamics and reasons for increased viral load in RA patients are not well understood. It most likely reflects that more cells retain latent virus either through the initial infection or subsequent reactivations with return to latency. The measured increase in virus likely reflects a latent state as patients retain a strong T cell and antibody response to the virus. Lytic infection from reactivation would be quickly controlled. In our work, we observe no reactivation of γHV68 during induction or maintenance of CIA here and in our previous EAE work. The act of maintaining latency is sufficient to induce heightened disease. Likewise, in our EAE work, we have observed that acute infection protects/delays from disease induction.

Line 47-50: for a better reading flow this should be moved after the sentence in line 42.

Thank you for this helpful suggestion, we have made the suggested change (line 45).

Line 52-55: references should be added.

Added (line 55).

Line 55: was this examined during an active or chronic EBV infection?

Unfortunately, the referenced manuscript does not examine EBV infection at this level. We agree that this would be helpful to know.

Line 64/65: this sentence is vague – perhaps 'enhancement' could be clarified.

Thank you for pointing this out – we have reworded and added additional information (line 71).

Line 69: does this strain reactive regularly? If yes, how did authors ensure in their experiments that the used EBV strain was not lytic during the onset of symptoms or tissue collection 56 days post-latency?

As described above, our group has previously shown that γHV68 does not reactivate following adjuvant. We have added these sentences to the text to further clarify: “Additionally, we find that inducing CIA in γHV68-infected mice does not impact viral load (Figure 1—figure supplement 1I), indicating that γHV68 is not reactivating. These findings are in line with our previous work showing that latent γHV68 infection enhances EAE without influencing autoantibody levels or reactivating γHV68(18)” (line 159).

Line 80: typo in C57BI/6 mice.

Corrected, thank you (line 102).

Line 83: given that the author's experiments do not truly uncover a mechanism by which EBV modulates RA disease, this sentence should be rephrased.

Thank you, we have rephrased this sentence, which now reads “We have utilized γHV68 infection and CIA induction to investigate mechanism(s) by which EBV contributes to RA, in particular through the modulation of age-associated B cells” (line 104)

Line 86: typo – to the contribution of EBV.

Corrected, thank you (line 108).

Line 93-95: references should be provided.

Added references, thank you (line 116).

Results:Line 110: point scale is later on reported to range from 0-3.

Thank you for pointing this out. We have revised to increase clarity (line 138).

Line 115/116: reference(s) should be added.

Added reference, thank you (line 146).

Line 155: more detailed explanation why specifically IFNγ and IL-17 two cytokines were chosen. See e.g. line 211.

Addressed above.

Line 156: typo in fold-change 129

Corrected, thank you (line 202).

Line 157: authors report a trend towards a reduced relative expression of IL-17, while the p-value is clearly non-significant (p=0.24). Same for line 189 with a p-value of p=0.19. Accordingly, findings should be rephrased.

Thank you for pointing this out. We have added a caveat that “the sample size is low due to the difficulty of obtaining these samples” (line 206) and have clarified that the decreased trend is non-significant.

Line 242: how does this relate to findings from reference 3 that RA patients have higher EBV loads, indicating that it is in a lytic stage?

In Balandraud et al., the researchers measured EBV DNA and this does not indicate replicating or lytic virus, just overall more virus. Initial EBV infections occur in childhood to early adulthood, where EBV latency is established and this is well before the onset of RA.

Line 267: not clear what is meant with viral enhancement?

We have reworded to increase clarity. Now reads: “...we investigated the role of ABCs in facilitating γHV68-exacerbation of CIA” (line 337).

Line 270: giving the non-significant p-values shown in Figure 5 B (p=0.077 and p=0.16), the total number of ABCs is not increased. Accordingly, findings and conclusions should be carefully rewritten.

Thank you for pointing this out. After changing our statistical tests to account for significantly different SDs, we find that there is a significant increase in total number of ABCs (Figure 5B).

Line 283/284: this sentence is vague.

Thank you for identifying this. We have reworded this sentence as “Further, fewer splenic ABCs in γHV68-CIA mice express inhibitory receptors CTLA4, PDL1, and PD1 (Figure 5D-F), and thus ABCs in CIA display a more regulatory phenotype than those in γHV68-CIA mice” (line 362).

Line 285-287: authors need to give more insights why certain parameters were examined.

Thank you for this observation. We have added this sentence: “We examined a series of markers previously shown to be expressed by ABCs, including cytokines IL10, IFNγ, and TNFα(26,48–50), an array of inhibitory receptors(26,30,36), maturity and memory markers IgD, IgM, and CD27(26,48), and MHCII(26,36,51)” (line 357).

Line 317: related Figures should be C and D, not D and E.

Corrected, thank you.

Line 317/318: results are overstated given the weak statistical support (p=0.07).

We have reworded this section and moved the immunological data to the supplements as per the following comment. Now reads: “We observe that the ablation of ABCs does not significantly alter the proportion of CD8, CD4, or Treg populations in the spleen during CIA or γHV68-CIA, nor the expression of IFNγ or IL17 (Figure 6—figure supplement 1). These results indicate that ABCs are a critical pathogenic population in γHV68-CIA though more work is needed to fully elucidate the mechanism by which ABCs are contributing to disease” (line 404).

Line 320/321: can this be added to the supplements?

Thank you for this suggestion, we have added an additional supplementary figure with the splenic T cell response in Ctrl and KO mice.

Figures:All p-values should be reported consistently (e.g. Figure 2A "ns", Figure 2B and C "p=0.24" and "p=0.22"). See also Figure 4.Stats are missing in Figure 3 D "ILN IFNγ of CD4" and Figure 3E, 6FFigure 6H (line 320) is missing completely

Corrected, thank you.

Only figure 6 is in color, while the rest is black/white

We have updated all figures to add color.

Material and Methods:Line 416: the authors should provide further details on the used strains, e.g. can/does yHV68 reactivate? When is the lytic strain ACRTA cleared?

Like EBV and other herpesvirus, γHV68 does reactivate from latency. We have added this to the list of attributes that γHV68 shares with EBV in the intro, which now reads: “γHV68 is a natural pathogen that is a well-established and widely-used murine model of EBV infection that shares an array of characteristics with human EBV infection, including latent persistence in B cells, viral reactivation from latency, a potent CD8 T cell response, and immune evasion tactics(19,20)” (line 82). As per ACRTA-γHV68, we have added the sentence: “In ACRTA-γHV68 the genes responsible for latency are deleted and a lytic gene, RTA, is constitutively expressed, resulting in clearance of the acute virus by day 14(48)” (line 287).

Reviewer #3 (Recommendations for the authors):[…] I have several comments/revisions/corrections and questions regarding the data:– Line 70: missing space between the letter "a" and the abbreviation "gHV68"

Thank you, corrected (line 102).

– Line 80: typo is the mouse strain – written "C7Bl/6" and should be "C57Bl/6"

Thank you, corrected (line 122).

– Line 86:- "The role of B cells the contribution of EBV to RA is intriguing…", correct the grammar.

Corrected, thank you (line 128).

– Line 94: In line 63 the abbreviation "lymphocytic choriomeningitis virus (LCMV)" was defined so when mentioning it the second time it should state the abbreviation and not the full name.

Thank you, corrected (line 116).

– Line 146: The authors state that "Synovial cells were not collected from naïve or gHV68-infected mice without CIA because we would not expect there to be sufficient infiltration of immune cells for analysis". On what basis do the author state this? Was it established that immune cells do not infiltrate the synovium w/o CIA?

We have attempted to collect synovium-infiltrating cells from non-inflamed joints (naïve and γHV68-infected mice) and were unable to secure enough sample for analysis.

– The author state (line 238) "… that ACRTA-gHV68 CIA mice display a similar clinical and immunological profile to uninfected CIA mice". However, in figure 4 they compare the ACRTA-gHV68 CIA group to gHV68 CIA and not to CIA only group. It would be important to see that there is no significant difference between the CIA and ACRTA-gHV68 CIA group to support their statement.

Thank you for this suggestion, we have added comparison of CIA and ACRTA-γHV68 CIA to Figure 4.

– Line 156 – fold chage -> fold change

Thank you, typo corrected (line 202).

– Line 157: The author describes an increase in IFNγ and decrease in IL-17 however, there is no statistical significance using the Mann-Whitney t-test. Although they describe a fold increase, they do not comment on the high p values.

Thank you for pointing this out. We have updated the p-values by using the Mann-Whitney test and have added that “the sample size is low due to the difficulty of obtaining these samples” (line 201).

– Line 164: Figure 2A – it is not clear how the numbers in the figure were calculated – % out of CD45+CD3+ cells? Moreover, not clear to what does the description "…filled circles) and γHV68-CIA mice (filled squares)" relates to in the figure legend?

Thank you for your comment and we apologize for the confusion. To clarify this, we have updated the figure description, which now reads: “(**A**) Representative flow cytometry plots of synovial fluid (SF) CD8^+^ and CD4^+^ T cells. Previously gated on lymphocytes, singlets, live cells, and CD45^+^CD3^+^ cells. Flow cytometry plots (concatenated samples) and graphs of total numbers (y-axis) of CD45^+^CD3^+^CD8^+^ and CD45^+^CD3^+^CD4^+^ T cells in uninfected mice with CIA (filled circles) and γHV68-CIA mice (filled squares)” (line 214).

– Line 188: The increase of CD8^+^ and decrease of FoXP3+ in the spleen of gHV68-CIA mice vs CIA mice is compared to a similar observation in the circulation of humans. Does this mean that the education process of T cells in the spleen is mirrored to the circulation? It would be interesting to check the same in the mouse circulation? (it is possible to recover PBMCs from mice ).

We agree that this is interesting and we have begun to address this in mice and examine their PBMCs.

– Line 189: " In the ILNs of gHV68-CIA mice, we observe a trend of the deceased relative proportion of regulatory T cells and…" – from figure 3A it does not seem that there is a decrease in Treg in the ILN of gHV68-CIA mice. Could the authors clarify this point?

Thank you for pointing this out. We have clarified this sentence, which now reads: “In the ILNs of γHV68-CIA mice, we observe a nonsignificant trend of decreased CD8^+^ and CD4^+^ T cells relative proportions, indicating potential T cell egress from the ILNs during disease, and find that the proportion of regulatory T cells is unchanged between CIA and γHV68-CIA mice (Figure 3A-B, Figure 3—figure supplement 1E-G)” (line 241).

– Line 228: When the authors write resembling uninfected CIA mice – did they check the significance of CIA onset (figure 4B) between CIA and ACRTA-γHV68 CIA ?

Yes, we have added that comparison, and find that there is not a difference in day of onset between CIA and ACRTA-γHV68 CIA (Figure 4B).

– Most of the comparisons made (as presented in figure 4) compare the ACRTA-γHV68 CIA mice to γHV68 CIA mice – my question is – does the infection with the mutated γHV68 affect these parameters in comparison to CIA mice?

We have added this comparison to Figure 4C-I and find no differences between ACRTA-MHV68 and CIA.

Reference:

Casiraghi C, Shanina I, Cho S, Freeman ML, Blackman MA, Horwitz MS (2012) Gammaherpesvirus Latency Accentuates EAE Pathogenesis: Relevance to Epstein-Barr Virus and Multiple Sclerosis. PLoS Pathog 8(5): e1002715. https://doi.org/10.1371/journal.ppat.1002715